# Instilling an Active Mind in Avatars via Cognitive Simulation

**Jianwen Jiang**[\*†]**, Weihong Zeng**[\*]**, Zerong Zheng**[\*]**, Jiaqi Yang**[\*]**, Chao Liang**[\*]**,
Wang Liao**[\*]**, Han Liang**[\*]**, Weifeng Chen, Xing Wang, Yuan Zhang, Mingyuan Gao**
ByteDance
jianwen.alan@gmail.com

## Abstract

Current video avatar models can generate fluid animations but struggle to capture a character's authentic essence, primarily synchronizing motion with low-level audio cues instead of understanding higher-level semantics like emotion or intent. To bridge this gap, we propose a novel framework for generating character animations that are not only physically plausible but also semantically rich and expressive. Our model is built on two technical innovations. First, we employ Multimodal Large Language Models to generate a structured textual representation from input conditions, providing high-level semantic guidance for creating contextually and emotionally resonant actions. Second, to ensure robust fusion of multimodal signals, we introduce a specialized Multimodal Diffusion Transformer architecture featuring a novel Pseudo Last Frame design. This allows our model to accurately interpret the joint semantics of audio, images and text, generating motions that are deeply coherent with the overall context. Comprehensive experiments validate the superiority of our method, which achieves compelling results in lip-sync accuracy, video quality, motion naturalness, and semantic consistency. The approach also shows strong generalization to challenging scenarios, including multi-person and non-human subjects. **Project page is linked here**.

## 1 Introduction

> "System 1 operates automatically and quickly, with little or no effort and no sense of voluntary control. System 2 allocates attention to the effortful mental activities that demand it, including complex computations."
>
> — Daniel Kahneman, *Thinking, Fast and Slow*

The field of video avatars (He et al., 2023; Tian et al., 2025c; Xu et al., 2024a; Wang et al., 2024a; Chen et al., 2024c; Xu et al., 2024b; Stypulkowski et al., 2024; Jiang et al., 2025; Lin et al., 2025a; Gan et al., 2025; Kong et al., 2025; Wang et al., 2025b; Hu, 2024b; Lin et al., 2025b; Qiu et al., 2025) aims to synthesize realistic character videos from driving signals, with the goal of creating lifelike digital humans capable of reasoned action and authentic emotion. Recent years have seen rapid progress, evolving from early lip-sync (Jiang et al., 2024; Zhang et al., 2023; Wang et al., 2021; Zhao & Zhang, 2022; Siarohin et al., 2019; 2021) and portrait animation (Jiang et al., 2025; Tian et al., 2025c; Zhong et al., 2024; Xu et al., 2024b;a; Cui et al., 2024) to half-body (Lin et al., 2025a; Tian et al., 2025a) and full-body generation (Lin et al., 2025b). As the scope of generation expands, so does the expectation for models to move beyond mere physical likeness and capture a character's authentic essence, their underlying personality, emotion, and intent, as shown in Figure 1.

A recent wave of audio-driven methods based on Diffusion Transformers (DiT) (Peebles & Xie, 2023; Esser et al., 2024b; Seawead et al., 2025; Kong et al., 2024; Wang et al., 2025a) can generate human motion synchronized with audio (Lin et al., 2025b; Kong et al., 2025; Gan et al., 2025; Wang et al., 2025b; Qiu et al., 2025). However, these models typically learn only low-level correlations, resulting in accurate lip movements but simple, repetitive gestures. Their outputs lack the

---

[\*]Equal contribution.
[†]Project Lead and Corresponding author.

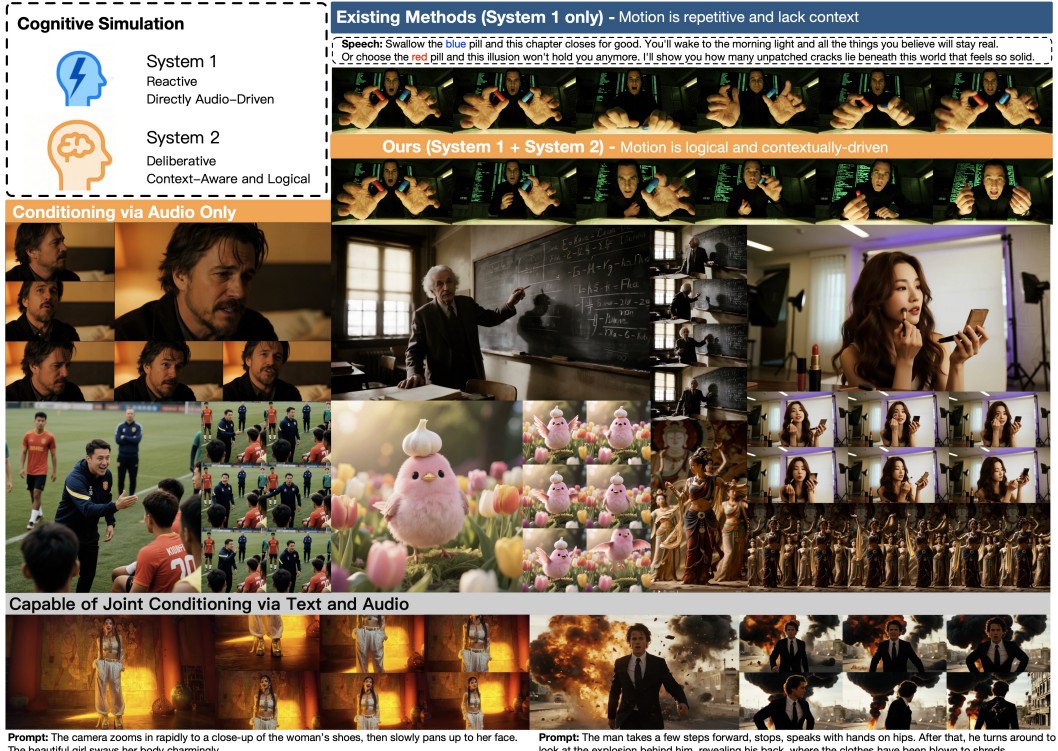

Figure 1: **Simulating a Mind for Avatars.** We model avatar behavior using dual-process theory, distinguishing between reactive (System 1) and deliberative (System 2) cognition. **Top Right:** Conventional methods, analogous to System 1, excel at reactive tasks like lip-sync but produce repetitive, non-contextual motions. **Top Left & Bottom:** In contrast, our framework simulates both systems, integrating System 2's high-level reasoning to generate diverse, context-aware behaviors that are semantically aligned with the audio and text.

contextual appropriateness and plausibility of authentic human behavior, revealing a significant gap between current capabilities and the goal of creating convincing avatars. We attribute this gap to a failure to model higher-order reasoning. Drawing inspiration from the dual-process theory of human cognition (Kahneman, 2011; Kahneman & Tversky, 2013), which distinguishes between fast, intuitive "System 1" thinking and slow, deliberative "System 2" reasoning, we observe that existing models operate mainly at the level of System 1. They excel at reactive mappings, like audio-to-lip-movement, but cannot perform the goal-oriented, contextual reasoning characteristic of System 2. We argue that the path forward lies in emulating both systems. To this end, we propose leveraging the powerful reasoning capabilities (Wei et al., 2022; Wang et al.; Yao et al., 2023; Schick et al., 2023; Park et al., 2023; Yuan et al., 2024; Wu et al., 2025b) of Multimodal Large Language Models (Team et al., 2023; OpenAI, 2024) to explicitly simulate the deliberative processes of System 2.

However, integrating MLLM-generated textual guidance into an avatar framework is non-trivial. The text, which articulates high-level reasoning, introduces a new modality that can conflict with existing signals like audio (for rhythm) and a reference image (for identity). Naive fusion can lead to modal interference, where, for example, low-level audio cues might disrupt high-level, text-guided semantic actions, and reference image conditioning could alter their motion magnitude. Therefore, a novel architecture is required to effectively manage these interdependencies and mitigate conflicts, enabling the simultaneous simulation of both System 1 and System 2.

Motivated by this analysis, we propose a Multimodal DiT framework with two key designs. First, an MLLM-powered agent generates a high-level, logically coherent "System 2" signal by reasoning over multimodal inputs.* Second, we introduce a specialized MMDiT architecture and training

---

*We employ the "System 1" and "System 2" dichotomy as a functional analogy. Our goal is not a neuro-scientific replication, but to adopt the interplay between reactive and deliberative processes to improve avatar coherence. For conciseness, we also use these terms to refer to the corresponding modules in our framework.

methodology to fuse these inputs and mitigate interference. This architecture incorporates a novel pseudo-last-frame strategy that preserves identity without constraining dynamic, content-driven motion, by leveraging the model's temporal extrapolation capabilities to maintain identity, instead of directly conditioning on the reference image during training. Our main contributions are as follows:

**A New Perspective on Avatar Modeling.** We are the first to frame the video avatar problem through the cognitive science lens of System 1 and System 2, identifying the limitations of current models and proposing a holistic approach that models both.

**A Framework for Dual-System Simulation.** We propose a novel framework featuring MLLM-based agents for deliberative "System 2" guidance and a specialized MMDiT architecture with a pseudo-last-frame strategy to synergistically fuse this guidance with reactive "System 1" signals, resolving critical modal conflicts.

**State-of-the-Art Performance and Generalization.** Our method achieves highly competitive results on multiple benchmarks and is significantly preferred in user studies for its contextual naturalness. Its versatility is further demonstrated by its successful extension to complex multi-person and non-human scenarios.

## 2    RELATED WORK

**Video Generation.** The field of video generation has rapidly advanced, largely building on the success of diffusion models in visual synthesis (Ho et al., 2020; Song et al., 2020). Current approaches can be broadly categorized by their architecture. The first category adapts pre-trained text-to-image U-Nets (Esser et al., 2024a; Chen et al., 2024a) by inserting temporal modules and fine-tuning on video data (Guo et al., 2023; Wang et al., 2023b). While leveraging powerful image priors, these methods can be constrained by their original image-centric design. The second category employs Diffusion Transformer (DiT) architectures (Brooks et al., 2024; Yang et al., 2024; Zheng et al., 2024; Kong et al., 2024; Ma et al., 2025; Polyak et al., 2024; Chen et al., 2024b; Menapace et al., 2024), which treat video as a sequence of spatiotemporal patches. This unified approach has demonstrated superior scalability and flexibility, enabling high-resolution video generation with variable durations. A third, emerging direction involves integrating Large Language Models (LLMs) to enhance logical coherence and narrative structure. While this approach is still nascent in video generation, it has shown great promise in the image domain (Yu et al., 2023b; Koh et al., 2023; Pan et al., 2025; Wu et al., 2025a; 2024; Shi et al., 2024) and holds significant potential for our field.

**Video Avatars.** Video avatar models aim to create realistic human videos from driving signals, with audio being a primary focus of recent research. These audio-driven animation methods must tackle the dual challenges of motion generation and rendering. A common paradigm is a two-stage pipeline: first, a model translates audio into an intermediate motion representation (e.g., keypoints, meshes, or tokens) (Zhuang et al., 2024; Meng et al., 2025; Deng et al., 2025; Wei et al., 2024; Hogue et al., 2024; Tian et al., 2025b); then, a rendering model synthesizes the final video. This rendering stage, which animates a character from an explicit motion sequence, constitutes an independent research area known as pose-driven animation (Zhang et al., 2024; Shao et al., 2024; Chang et al., 2023; Tu et al., 2024; Wang et al., 2024b; Karras et al., 2023; Xu et al., 2024c), where the primary focus is on rendering fidelity. In contrast to these two-stage methods, recent end-to-end approaches directly generate video from audio for better synchronization (Lin et al., 2025a;b; Wang et al., 2025c; Liang et al., 2025). Despite their architectural diversity, all these audio-driven methods fundamentally treat motion generation as a direct, reactive mapping. This process lacks an explicit cognitive phase of planning and reasoning, which our work introduces to generate more plausible and intelligent behaviors.

**LLMs for Cognitive Simulation and Control.** Large Language Models (LLMs) have evolved from foundational models (Brown et al., 2020; Ouyang et al., 2022; Achiam et al., 2023) to powerful cognitive engines with advanced reasoning (Hurst et al., 2024; Jaech et al., 2024; Guo et al., 2025a), unlocked by prompting techniques such as Chain-of-Thought (Wei et al., 2022; Wang et al.; Yao et al., 2023) and other applications (Liang et al., 2024; Kannan et al., 2024; Qu et al., 2023). This has empowered autonomous agents to strategize and simulate believable human behaviors (Schick et al., 2023; Hong et al., 2023; Wang et al., 2023a; Park et al., 2023). Beyond standalone agents, LLM reasoning is also used to steer generative models, acting as a universal planner for controllable

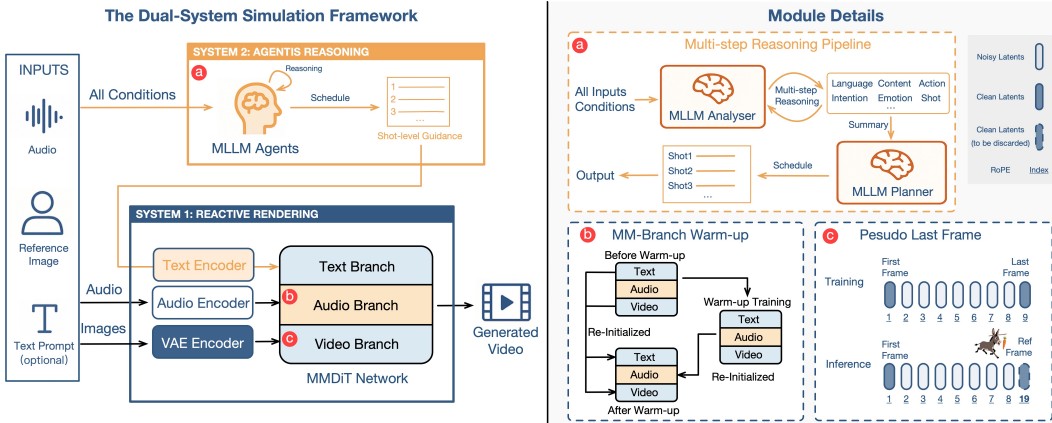

Figure 2: **The Dual-System Simulation Framework.** Our framework models avatar behavior by integrating a deliberative System 2 for planning with a reactive System 1 for synthesis. **Left: Overall Pipeline.** An MLLM-based System 2 reasons over multimodal inputs (audio, image, text) to generate a high-level "schedule". This schedule guides the System 1 MMDiT, which synthesizes the final video by fusing information through dedicated text, audio and video branches. **Right: Key Components.** (a) The System 2 reasoning pipeline, comprising an MLLM Analyser and Planner. (b, c) Our proposed MM-Branch Warm-up and Pseudo Last Frame strategies, designed to mitigate modal conflicts.

image and video synthesis (Brooks et al., 2023; Pan et al., 2025; Hu et al., 2024a; Yuan et al., 2024; Li et al., 2024; Wu et al., 2025b). However, applying the reasoning capabilities of these powerful models to generate fine-grained, intelligent avatar behavior remains a largely unexplored area which our work directly addresses.

## 3 APPROACH

### 3.1 OVERVIEW

Our goal is to generate character animations that are both visually realistic and logically coherent with multimodal inputs. To this end, we introduce a framework that simulates both reactive (System 1) and deliberative (System 2) cognitive processes, as illustrated in Figure 2. The core of our model is a Diffusion Transformer (DiT) backbone (Seawead et al., 2025; Gao et al., 2025; Esser et al., 2024b; Peebles & Xie, 2023), pre-trained on general video tasks to acquire foundational synthesis capabilities. We enhance this base model with two critical designs. To simulate System 2 (Sec. 3.2), we employ MLLM-based agents to reason over the input context and generate high-level semantic guidance for deliberative control. To simulate System 1 (Sec. 3.3), a specialized Multimodal DiT (MMDiT) architecture (Esser et al., 2024b) fuses this guidance with reactive signals like audio. To mitigate modal conflicts within the MMDiT, we introduce a pseudo-last-frame strategy that prevents the static reference image from interfering with dynamic motion generation.

Following common practices for simplicity (Lin et al., 2025b; Gan et al., 2025; Kong et al., 2025), our framework operates in the latent space of a pre-trained 3D VAE (Yu et al., 2023a) and is trained with a flow matching objective (Liu et al., 2022). To support long-form video synthesis, the model can operate autoregressively by conditioning new segments on the final frames of the previous one (Stypulkowski et al., 2024). We omit further discussion of these standard components to focus on our core contributions.

### 3.2 AGENTIC REASONING FOR DELIBERATIVE CONTROL

**Agentic Reasoning for Deliberative Guidance.** To simulate the deliberative nature of "System 2", our agentic reasoning module processes multimodal inputs to generate high-level, logically coherent guidance. The module takes the character's reference image, the audio clip, and an optional text

prompt describing the desired behavior. It outputs this guidance as reasoning text, the explicit textual output generated by the MLLM agents in response to our designed step-by-step guided probing, which directly conditions the synthesis model.

**Multi-Step Reasoning Pipeline.** As shown in Figure 2 (top-right), this guidance is generated by a two-stage MLLM pipeline. First, an Analyzer MLLM receives the reference image, its caption, the audio, and the user prompt. Guided by step-by-step probing prompts, the model infers the character's speech content, emotional state, and intent. These insights are then consolidated into a structured JSON object. This output is then passed to a Planner MLLM, which uses this context to devise a detailed action plan. This plan is structured as a sequence of shots, with each shot defining the character's expressions and actions for a single generation pass. This collaborative process yields a comprehensive motion schedule that ensures a coherent character persona across the entire video. Details and examples are provided in Appendix D.

**Framework Extensibility.** The flexibility of our agentic framework also enables various design explorations. For instance, to enhance long-form coherence, the Planner can incorporate a reflective re-planning step to correct for semantic drift during synthesis. We also explored an alternative conditioning method using reasoning-infused audio latents. While we leave a full investigation of these extensions to future work, our primary experiments focus on the straightforward reasoning text approach for its proven robustness and effectiveness. We provide preliminary examples of these explorations in Appendix E to illustrate the broader potential of our framework. This chosen design enables our agent to formulate a global, coherent plan, integrating deliberative "System 2" reasoning to provide thoughtful, top-down guidance absent in purely reactive "System 1" methods.

### 3.3 REACTIVE RENDERING VIA MULTIMODAL DIFFUSION

In this section, we detail how our diffusion model synthesizes the final video. It synergistically fuses high-level guidance from the System 2 agents (represented as text) with low-level, reactive signals from the audio input.

**Rethinking Reference Image Conditioning.** A critical input in video avatar models is the reference image, which serves two distinct purposes: first, providing an initial frame as a conditioning prefix for the generated sequence, and second, maintaining identity consistency. While the former is a necessary function, the latter (using a static reference image to enforce identity) is problematic. Prior methods, whether using dedicated networks (Hu, 2024a; Tian et al., 2025c; Zhu et al., 2023) or parameter reuse (Lin et al., 2025b; Kong et al., 2025), typically condition the model on a reference image sampled from the training video. This training strategy teaches the model a spurious correlation: as illustrated in Figure 3, it learns that the reference image must appear literally within the generated sequence. We argue this is a critical artifact that severely restricts motion dynamics and conflicts with other driving signals. The root cause is that the reference image is an artificial construct, not a condition native to the video data itself. This leads to a training dilemma regarding the semantic distance of the reference image. Sampling a semantically "close" reference (e.g., from the same clip) creates the static artifact, while sampling a semantically "distant" one can teach the model to exhibit excessive, identity-altering variation.

Our solution is to discard the reference image entirely during training and introduce a novel guidance mechanism. As shown in Figure 2 (bottom-right), we instead probabilistically condition the model on the GT first and last frames of the video clip, both native signals, each with a dropout probability of 0.1. During inference, we repurpose this mechanism by placing the user's reference image in the last frame's position, creating a "pseudo-last-frame". Crucially, we shift its positional encoding, RoPE (Su et al., 2024), by assigning it a positional index corresponding to a fixed temporal distance beyond the final generated frame. This pseudo frame functions as a "carrot on a stick": it guides the model toward the target identity without ever forcing it to replicate the static image, which is discarded after synthesis. As our experiments show, this approach eliminates training artifacts and mitigates autoregressive error, achieving a superior trade-off between motion dynamics and identity stability.

**Symmetric Fusion and Modality Warm-Up.** Having established data-native conditions, we now address their joint modeling. We adopt an MMDiT backbone but introduce a dedicated audio branch, architecturally symmetric to the video and text branches. Instead of using cross-attention, all three modalities are fused within each transformer block by concatenating their tokens and applying a sin-

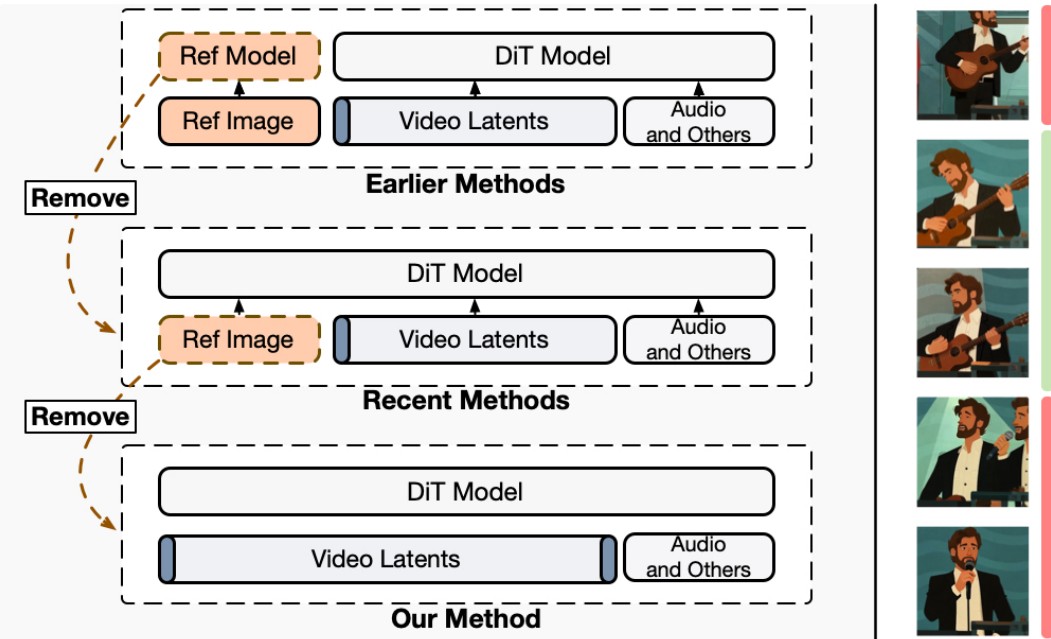

Figure 3: **The Reference Conditioning Dilemma. Left:** The evolution of reference conditioning methods towards simpler designs. **Right:** The core sampling trade-off. Sampling **within-clip** ensures high relevance but teaches the model to produce static results. Conversely, sampling **outside-clip** introduces low-relevance signals, causing the model to generate identity or content inconsistencies.

gle shared multi-head self-attention mechanism. This symmetric design enables true joint modeling, as tokens from all modalities mutually attend to one another, allowing for iterative refinement and deep semantic alignment.

However, this architecture introduces a critical training challenge: naive joint training causes the model to over-rely on the dense audio signal, which washes out text guidance and disrupts the pre-trained video branch's patterns, degrading overall synthesis capability. To resolve this, we propose a two-stage warm-up strategy. In stage 1, we train the full three-branch model jointly, forcing the model to learn an optimal division of labor where the audio branch specializes in its core competencies (e.g., lip sync, speech mannerisms). In stage 2, we initialize the text and video branches with their original weights and the audio branch with its specialized weights from stage 1, then fine-tune the entire model. This strategy provides each branch with a strong prior, mitigating modality conflict and preserving each input's distinct conditioning power.

Ultimately, our redesigned reference conditioning and symmetric audio fusion enable the model to effectively execute the deliberative plan of System 2, translating high-level guidance while maintaining the reactive fidelity of System 1.

## 4 EXPERIMENTS

### 4.1 EXPERIMENTAL SETUP

**Implementation and Training.** Our model, built upon the MMDiT architecture, generates 120-frame clips at 24 fps. For efficiency, most ablation studies are conducted at a 480p resolution. For comparison with prior work, these clips are then upscaled to 720p to match common evaluation setting. The model underwent a three-stage training process: an audio branch warm-up, a main phase on 15,000 hours of video, and a final fine-tuning stage on a 100-hour high-quality subset. Further implementation and training details are provided in the Appendix B.

**Evaluation Datasets and Metrics.** To rigorously test generalization, we constructed two challenging custom benchmarks: a diverse single-subject set (150 cases) and a multi-subject set (57 cases),

Table 1: **Ablation studies on our proposed framework.**

| Method | IQA ↑ | ASE ↑ | Sync-C ↑ | HKC ↑ | HKV ↑ |
|---|---|---|---|---|---|
| *Ablation on Agentic Reasoning* | | | | | |
| Ours w/o Multi-Step Reasoning | 4.795 | 3.901 | 3.853 | 0.576 | 157.638 |
| Ours w/o Analyzer | 4.793 | 3.910 | 4.278 | 0.572 | 148.381 |
| Ours w/o Reasoning (System 1 Only) | 4.784 | 3.885 | 3.507 | 0.544 | 122.376 |
| *Ablation on Conditioning Modules* | | | | | |
| Ours w/ Cross-Attention | 4.745 | 3.856 | 3.263 | 0.558 | 116.317 |
| Ours w/o MM-Warmup | 4.752 | 3.866 | 3.993 | 0.549 | 164.080 |
| Ours w/ Ref. Image | 4.772 | 3.896 | 3.982 | 0.559 | 160.889 |
| Ours w/o Ref. & Pseudo Frame | 4.682 | 3.878 | 4.141 | 0.564 | 160.986 |
| **Ours (Full Model)** | 4.790 | 3.901 | 4.087 | 0.571 | 168.912 |

Table 2: **Pairwise subjective ablation study and component comparison.** We report Lip-sync Inconsistency (LSI), Motion Unnaturalness (MU), Image Distortion (ID), and an overall Good/Same/Bad preference score (GSB). Lower is better for LSI, MU, and ID.

(a) Ablation on reasoning.

| Method | LSI↓ | MU↓ | ID↓ | GSB↑ |
|---|---|---|---|---|
| Ours (w/o Reasoning) | 0.12 | 0.58 | 0.11 | −0.29 |
| Ours (Full Model) | 0.12 | 0.37 | 0.04 | +0.29 |

(b) Comparison of conditioning.

| Conditioning Method | LSI↓ | MU↓ | ID↓ | GSB↑ |
|---|---|---|---|---|
| OmniHuman-1 | 0.21 | 0.39 | 0.17 | −0.23 |
| Ours (Proposed) | 0.03 | 0.25 | 0.07 | +0.23 |

(c) GSB Score Comparisons

| GSB Comparisons | TA ↑ | Mot ↑ | VQ ↑ |
|---|---|---|---|
| Ours vs. Base Model | -0.02 | +0.18 | +0.14 |

both featuring a wide range of characters and complex audio. For fair comparison with prior work, we also evaluate on the CelebV-HQ (Zhu et al., 2022) and CyberHost (Lin et al., 2025a) test sets.

Our evaluation protocol includes both objective and subjective metrics. We measure image quality (FID, IQA), video quality (FVD), lip-sync (Sync-C), and gesture dynamics (HKC, HKV). Crucially, to assess perceptual quality, we conducted user studies with 40 participants, using pairwise comparisons (GSB score) and flaw identification (LSI, MU, ID) to capture nuances that objective metrics miss. A detailed description of our datasets and evaluation protocols can be found in the Appendix C.

## 4.2 ABLATION STUDIES

We conduct a series of ablation studies to validate the contributions of our proposed components, using a custom single-subject test set of 150 video clips. Our analysis systematically isolates the impact of two key elements: (1) the agentic reasoning module and (2) our proposed conditioning architecture. For a comprehensive assessment, we employ both quantitative metrics and subjective user studies to evaluate performance, perceptual quality, and user preference.

**Effectiveness of Agentic Reasoning.** We analyze our Agentic Reasoning module by progressively ablating its components: first removing multi-step reasoning, then the Analyzer, and finally the full module for a "System 1 only" baseline. As shown in Table 1, standard metrics like IQA and Sync-C show minimal variation. This is expected, as these metrics measure low-level fidelity, are largely saturated, and are insensitive to higher-level semantics. A more telling trend emerges from Hand Keypoint Variance (HKV), which progressively decreases as reasoning is removed, indicating that the generated motion becomes more static and less expressive. This directly demonstrates the value of our reasoning module.

To assess the module's semantic impact, we conducted subjective evaluations. As shown in Table 2 (a), our full model shows a substantial advantage. Agentic reasoning yields a 29% gain in GSB and a significant reduction in perceived motion unnaturalness (MU), while maintaining strong lip-sync (LSI) and image quality (ID). These findings confirm our reasoning module enhances motion plausibility and naturalness, qualities not captured by standard objective metrics. **This is further corroborated by an MLLM-based analysis** (Appendix D.2) confirming our method's superior contextual coherence. **We strongly encourage viewing the supplementary material and project page**, which showcase compelling visual results unattainable with existing "System 1" models.

Table 3: **Comparison with existing methods on multi-person animation.** We report quantitative metrics and pairwise subjective evaluation results, including Driving Accuracy (DA), Lip-sync Inconsistency (LSI), Motion Unnaturalness (MU) and an overall user preference score derived from a Good/Same/Bad (GSB) evaluation.

| Method | Subjective Evaluation | | | | Quantitative Metrics | | | | |
|---|---|---|---|---|---|---|---|---|---|
| | DA↑ | LSI↓ | MU↓ | GSB↑ | IQA↑ | ASE↑ | Sync-D↓ | HKC↑ | HKV↑ |
| InterActHuman | - | - | - | - | 4.574 | 3.643 | 8.163 | 0.553 | 103.91 |
| Ours w/o Reasoning | 0.88 | 0.13 | 0.63 | -0.26 | 4.576 | 3.631 | 7.541 | 0.611 | 138.43 |
| **Ours (Full Model)** | 0.94 | 0.04 | 0.12 | +0.26 | 4.529 | 3.653 | 6.904 | 0.614 | 158.36 |

Table 4: **Quantitative comparison with audio-conditioned animation baselines. (Left)** Portrait animation on the CelebV-HQ test set. **(Right)** Full-body animation on the CyberHost test set.

| Method | IQA↑ | ASE↑ | Sync-C↑ | FID↓ | FVD↓ |
|---|---|---|---|---|---|
| SadTalker | 2.953 | 1.812 | 3.843 | 36.648 | 171.848 |
| Hallo | 3.505 | 2.262 | 4.130 | 35.961 | 53.992 |
| EchoMimic | 3.307 | 2.128 | 3.136 | 35.373 | 54.715 |
| Loopy | 3.780 | 2.492 | 4.849 | 33.204 | 49.153 |
| Hallo-3 | 3.451 | 2.257 | 3.933 | 38.481 | 42.125 |
| OmniHuman-1 | 3.875 | 2.656 | 5.199 | 31.435 | 46.393 |
| Ours | 3.817 | **2.663** | 5.053 | **31.320** | 45.771 |

| Method | IQA↑ | ASE↑ | Sync-C↑ | FID↓ | FVD↓ | HKC↑ | HKV↑ |
|---|---|---|---|---|---|---|---|
| Skyreel-A1 | 3.889 | 2.525 | 2.983 | 69.619 | 70.678 | 0.786 | 28.840 |
| FantasyTalking | 3.892 | 2.738 | 3.548 | 52.332 | 47.052 | 0.838 | 18.845 |
| OmniAvatar | 3.871 | 2.728 | 6.589 | 42.163 | 43.998 | 0.795 | 56.574 |
| MultiTalk | 3.822 | 2.681 | 6.868 | 37.308 | 32.783 | 0.817 | 62.753 |
| OmniHuman-1 | 4.142 | 3.024 | 7.443 | 31.641 | 27.031 | 0.898 | 47.561 |
| Ours | **4.144** | **3.030** | 7.243 | **31.160** | 27.642 | 0.875 | **72.113** |

**Effectiveness of Proposed Conditioning Modules.** We also ablate our core architectural designs in Tables 1 and 2, keeping the agentic reasoning module fixed. We ablate our core architectural designs by evaluating three key changes: replacing our MM-Attention-based audio conditioning with standard cross-attention, removing the MM-Warmup stage, and substituting our pseudo-last-frame strategy with conventional reference image attention. Our full model consistently leads in objective metrics, with superior HKC/HKV scores highlighting enhanced motion dynamics. Subjectively, our method also significantly outperforms a baseline using cross-attention for both audio and reference image conditioning (akin to OmniHuman-1 (Lin et al., 2025b)), showing clear advantages in lip-sync (LSI), motion naturalness (MU), and visual quality (ID). These results validate our unique conditioning architecture as a robust foundation for executing agentic plans. Further visual analysis in the appendix demonstrates how PLF maintains identity consistency during dynamic motion.

**Text-Conditioning Fidelity.** As a supplementary experiment, we verified that our multi-modal training does not degrade text-following capabilities. Under text-only conditioning, a user study (Table 2) confirms our model achieves on-par text alignment (TA) with the base model, while demonstrating significantly improved motion naturalness (MU) and visual quality (ID). This validates that our approach enhances generation quality without sacrificing core text-conditioning fidelity.

### 4.3 FURTHER EXPLORATION ON APPLICATIONS

**Generalization to Diverse and Multi-Person Scenarios.** Our model demonstrates robust generalization, enabled by a dual-system framework whose agentic reasoning provides context-aware motion guidance for non-human subjects and interprets conversational turn-taking (Figure 4). To further showcase this extensibility, we conducted a preliminary study on multi-person animation by incorporating a predicted speaker mask to guide the system (details in Appendix B). In a focused comparison (Table 3), our full model significantly outperforms baselines like InterActHuman (Wang et al., 2025c) and an ablation without reasoning. The observed improvements in gesture dynamics (HKC/HKV), lip-sync (Sync-D), and driving accuracy (DA) primarily validate the effectiveness of our reasoning component in enhancing coordination in such complex settings. These results underscore our method's broad applicability and potential for creating interactive avatars (see supplementary material for video results).

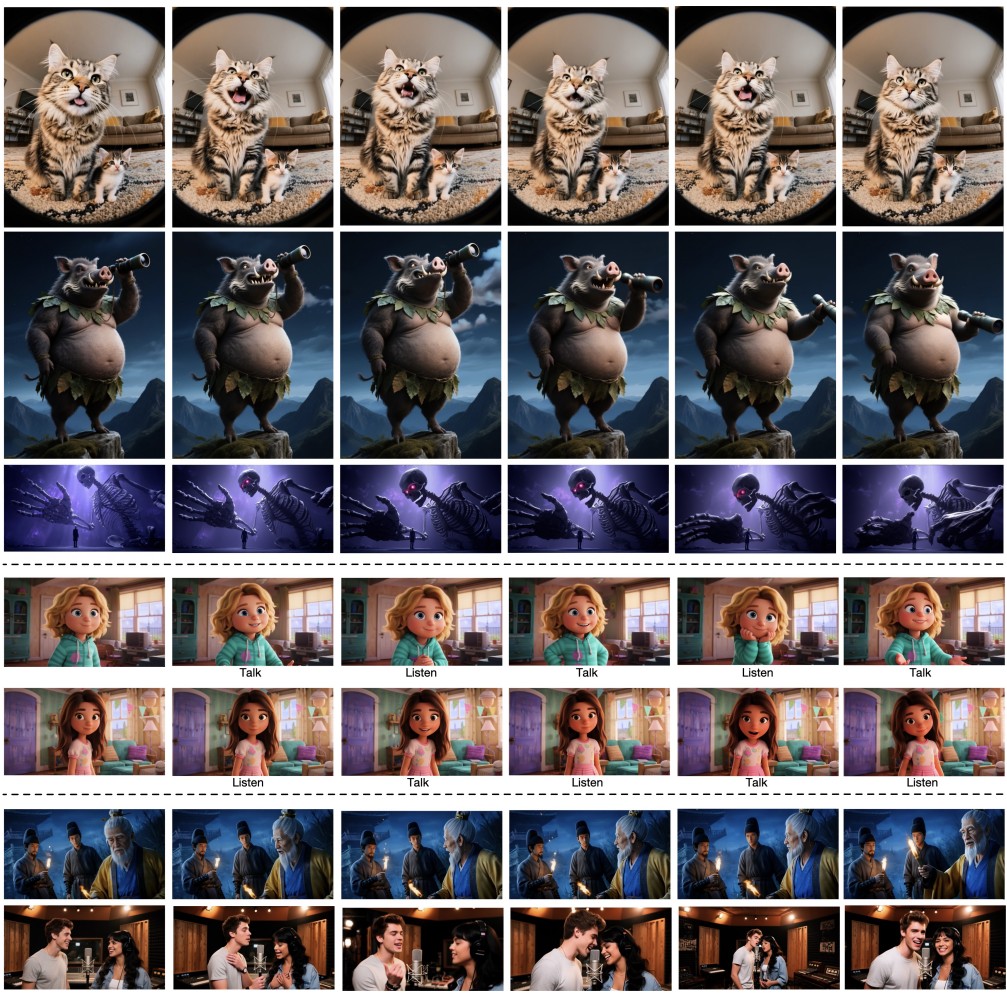

Figure 4: **Generalization and Multi-Person Results.** Our model generalizes to diverse non-human subjects (top rows), correctly interprets conversational turn-taking (fourth row), and generates coordinated behavior in multi-person scenes (bottom rows).

Figure 5: **Subjective User Preference Study.** We present results from two evaluation settings: (Left) a best-choice selection task comparing our method against academic baselines, and (Right) a GSB pairwise comparison against leading proprietary models.

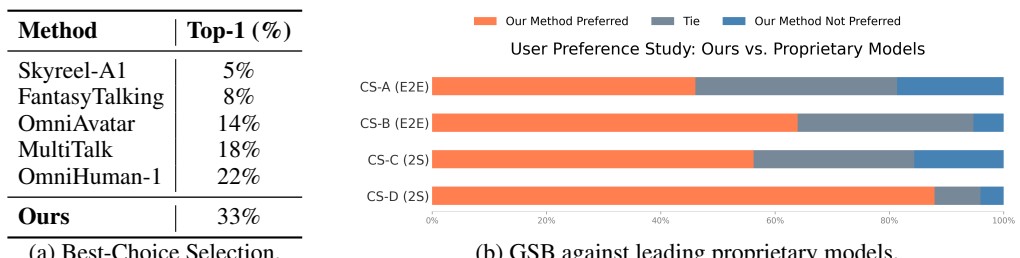

| Method | Top-1 (%) |
|---|---|
| Skyreel-A1 | 5% |
| FantasyTalking | 8% |
| OmniAvatar | 14% |
| MultiTalk | 18% |
| OmniHuman-1 | 22% |
| **Ours** | 33% |

(a) Best-Choice Selection.

(b) GSB against leading proprietary models.

## 4.4 COMPARISON AMONG RECENT METHODS

**Comparisons with State-of-the-Art Methods.** We conduct a comprehensive evaluation against leading academic baselines across two scenarios: portrait and full-body generation. For portraits, we compare on the CelebV-HQ test set against specialized and DiT-based methods, including SadTalker (Zhang et al., 2023), EchoMimic (Chen et al., 2024c), Hallo (Xu et al., 2024a),

Hallo3 (Cui et al., 2024), Loopy (Jiang et al., 2025), and OmniHuman-1 (Lin et al., 2025b). For the more challenging full-body synthesis task, we evaluate on the CyberHost test set against recent DiT models like Skyreels-A1 (Fei et al., 2025), FantasyTalking (Wang et al., 2025b), OmniAvatar (Gan et al., 2025), MultiTalk (Kong et al., 2025), and OmniHuman-1.

Quantitative results (Table 4) show our method consistently ranking among the top two. While on par with OmniHuman-1 in portraits, a setting where limited motion challenges objective metrics, our advantages become pronounced in the full-body setting (see also Table 2b and Figure 12). In this setting, our model excels at generating dynamic, large-scale movements (high HKV) while preserving local detail (competitive HKC). To assess perceptual quality, we conducted user studies against a top academic method (Figure 5a) and four leading, anonymized commercial systems (Figure 5b). In user studies, human evaluators consistently and significantly preferred our results overall, a preference we attribute to high-level qualities like contextual coherence that objective metrics fail to capture but are critical for perceptual realism.

## 5  CONCLUSION

Inspired by the dual-system theory of human cognition, we introduce a new paradigm for video avatars. We argue that existing methods, by simulating only reactive "System 1" thinking, fail to align motion with high-level intent, resulting in behaviors that lack contextual appropriateness and logical coherence. To address this, we propose a novel framework that models deliberative "System 2" processes via two innovations: MLLM-based agents for semantic planning and a specialized MMDiT for high-fidelity synthesis. Experimental results validate that our approach generates more expressive and logically coherent motions in both single- and multi-person scenarios, achieving leading performance on both subjective and objective metrics across multiple benchmarks. We hope this cognitive agency perspective offers the community a promising path toward creating video avatars that are not just visually realistic, but truly believable.

**Ethics Statement.** Our core contribution is a novel paradigm for video avatar generation. By simulating a dual-system cognitive framework, our model achieves a new level of expressive capability and logical coherence, moving beyond the limitations of single-process generation. While this advancement opens exciting possibilities for creative applications like AI-driven film production, we recognize the potential for misuse associated with highly realistic avatar technologies. To address these ethical concerns, we advocate for a robust framework of responsible deployment. Given its potentially powerful capabilities, this model requires strict access control to prevent potential abuse. Although current results may still bear subtle artifacts that can serve as an informal deterrent, proactive safeguards are essential. We strongly recommend the following measures: (1) applying prominent, visible watermarks to all generated content to clearly label it as AI-generated; (2) implementing filtering algorithms to reject inappropriate or malicious input prompts and to review output content; and (3) embedding traceable, invisible watermarks to ensure accountability and aid in source identification if misuse occurs. By integrating these safety protocols, we can help ensure that our technology fosters creativity while minimizing the risks of malicious applications such as fraud or disinformation.

**Reproducibility Statement.** To ensure the full reproducibility of our work, we provide comprehensive details regarding our methodology, data, implementation, and evaluation procedures across the main paper and its appendices. Specifically, we detail: (1) the core architecture of our agentic system and the MMDiT model in Section 3; (2) complete implementation details, including all hyperparameters and the multi-stage training procedure, in Appendix B; (3) our data curation pipeline, with specific filtering tools (e.g., PySceneDetect, Q-Align) and a statistical analysis of the dataset, also in Appendix B; and (4) our agentic system and evaluation framework, which specifies the MLLM models used (miniCPM-o and Seed-1.5-VL), instruction prompts, and the complete set of objective and subjective evaluation metrics in Appendices D and C. Finally, while we provide these details to ensure technical reproducibility, we reiterate our ethical recommendation that for responsible deployment, access to the trained models should be managed in a controlled environment to monitor usage and prevent potential misuse.

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

## A  APPENDIX

This appendix provides comprehensive supplementary material to the main paper, organized as follows:

- **Section B (Implementation and Training):** Details our model architecture, multi-person extension, training procedures, and data curation pipeline.

- **Section C (Evaluation Details):** Describes the novel datasets and the multi-faceted evaluation protocol, covering both objective and subjective metrics.

- **Section D (Agentic System Analysis):** Offers a deeper analysis of the reasoning system, with process demonstrations, MLLM specifications, and the full prompt for our MLLM-based evaluation.

- **Section E (Qualitative Results):** Presents visual comparisons against baselines and detailed ablation studies for key components (e.g., PLF, MM-Warmup, reflection).

- **Section F (System Latency):** Discusses the inference latency of the agentic system, framing it as a necessary trade-off for achieving higher-level coherence.

- **Section G (Statements):** Concludes with statements on ethical considerations, our use of Large Language Models (LLMs) in preparing the manuscript, and our efforts to ensure reproducibility.

- **Section H (Limitations):** Provides a discussion of the current limitations of our approach, including an analysis of failure cases, and outlines promising avenues for future work.

## B  IMPLEMENTATION AND TRAINING DETAILS

**Implementation Details.** Our model is based on the Multi-modal Diffusion Transformer (MMDiT) architecture, sharing a similar design with recent works (Lin et al., 2025b; Seawead et al., 2025; Gao et al., 2025; Kong et al., 2024), and was pre-trained on a large-scale text-video dataset. The architecture processes separate branches for video, text, and audio, which are subsequently fused using multi-modal attention layers. We introduce the audio branch by duplicating the pre-trained text branch for parameter initialization, with its input derived from audio features extracted by a Whisper (Radford et al., 2023) encoder. To maintain computational efficiency, each visual latent performs multi-modal attention with only its five temporally nearest audio and text tokens. Given that the audio and text token sequences are significantly shorter than the visual token sequence, this architectural extension introduces negligible computational overhead.

For most experiments, the model generates 120-frame clips at 24 fps and a 480p resolution. For visualization and comparison purposes, these outputs are upscaled to 720p using a separate, identically architected super-resolution model. Longer videos are produced autoregressively; specifically, for our pseudo-last-frame strategy, the final five frames of a generated clip serve as the initial conditioning frames for the subsequent clip, with a RoPE position shift of 30. A detailed analysis of this shift value is provided in the Appendix (see Figure 9). For optimization, we employed the AdamW optimizer with a learning rate of 5e-5, a global batch size of 256, and gradient clipping at a norm of 1.0. The training process was conducted in three phases: an audio branch warm-up ($\sim$18k GPU-hours, measured on an A100 equivalent), a main training phase ($\sim$43k GPU-hours), and a final fine-tuning phase on high-quality data ($\sim$6k GPU-hours). The MLLM Agent is designed to be pluggable, thanks to a flexible framework architecture that supports both local and API-based models. For this work, we selected miniCPM-o (Hu et al., 2024b) as the Analyser and Seed-1.5-VL (Guo et al., 2025b) as the Planner, based on their top-tier performance on relevant leaderboards.

**Implementation of Multi-Person Scenarios.** To enable multi-person animation, we extend our model with two key modifications. First, we condition the synthesis on a speaker-specific mask. This mask directs the injection of audio features exclusively to the corresponding masked regions during the multimodal attention process. Following the approach of InterActHuman (Wang et al., 2025c), we employ a lightweight, plug-and-play predictor to dynamically generate these masks. This ensures robust speaker tracking through movement and occlusion without altering the baseline single-person model. Second, we leverage our framework's inherent agent-based design by

augmenting the Planner. Specifically, the Planner is modified to accept this speaker mask as an additional input alongside the reference image. The mask is a binary image where pixels corresponding to the active speaker in the reference image are set to 255, while all other areas are 0. We provide this mask to the Planner with a supplementary instruction clarifying its role as a speaker indicator. With the rest of the reasoning pipeline remaining identical, this simple yet effective extension enables the model to generate logically consistent and coordinated actions for all individuals in the scene.

**Training Data Curation and Composition.** Our training set comprises 15,000 hours of video data, curated through a multi-stage pipeline inspired by prior works (Lin et al., 2025b; Cui et al., 2024) to ensure data quality and relevance. The filtering process involved the following steps:

1. **Temporal Segmentation and Filtering:** We first processed all videos for temporal consistency. We utilized **PySceneDetect** to segment videos at shot transitions, retaining only continuous clips with durations between 5 and 20 seconds. We then filtered for videos containing human subjects based on their captions.

2. **Visual and Motion Quality Enhancement:** We employed **PaddleOCR** to identify and remove clips with intrusive or changing subtitles. The visual fidelity and aesthetic appeal of each video were then assessed using **Q-align** (Wu et al., 2023), discarding any clips that fell below a predefined quality threshold. To ensure the motion was suitable for learning, we calculated optical flow with **Raft** (Teed & Deng, 2020) and filtered out videos exhibiting either excessively static or chaotic content.

3. **Audio-Visual Synchronization:** For our audio-driven generation task, we performed a crucial filtering step using **SyncNet** (Chung & Zisserman, 2017) to verify the synchronization between lip movements and the audio track, discarding out-of-sync examples. The resulting dataset was used for the main training phase. For samples with poor lip-audio correlation (constituting approx. 70% of the data), we discarded the audio and utilized them with audio-dropout during training.

4. **High-Quality Subset for Fine-Tuning:** For the final fine-tuning stage, we required a higher-quality selection. We curated the top 100 hours of data by ranking all clips based on a suite of quality metrics, including aesthetics (Q-align) and lip-sync accuracy (SyncNet).

To better understand the composition of our final dataset, we analyzed the distribution of a randomly sampled subset. In terms of character framing, the data primarily consists of closer shots: upper-body shots (from the chest up) constitute approximately 47% of the clips, followed by medium shots (from the hips up) at 30%. Full-body and long shots make up the remaining 15% and 8%, respectively. Regarding scene type, the dataset shows a notable preference for indoor environments (approx. 45%), while outdoor scenes with static, high-quality backgrounds represent 41%, and those with dynamic backgrounds constitute the final 14%.

## C  EVALUATION DETAILS

**Evaluation Datasets.** To rigorously assess our model, we recognized that existing DiT-based methods already perform well in standard human speaking scenarios. We therefore constructed two novel and highly challenging test sets to probe the limits of generalization beyond standard benchmarks.

- **Single-Subject Generalization Set:** A diverse set of 150 cases, featuring real-world human portraits, AIGC figures, anime characters, and animals. Each image was expertly paired with a corresponding audio track (e.g., speech, singing, theatrical performance) to create a demanding generalization test. To assess text-conditioning, experts also authored descriptive prompts for all 150 cases.

- **Multi-Subject Interaction Set:** A set of 57 cases with similar visual diversity to the single-subject set. Audio tracks were expert-paired to reflect multi-character interactions, enabling the evaluation of coordinated behaviors.

For fair comparison with prior work, we also adopted their experimental settings, using 100 videos from CelebV-HQ (Zhu et al., 2022) for the talking-head task and the CyberHost (Lin et al., 2025a) test set (269 videos, 119 identities) for full-body scenarios.

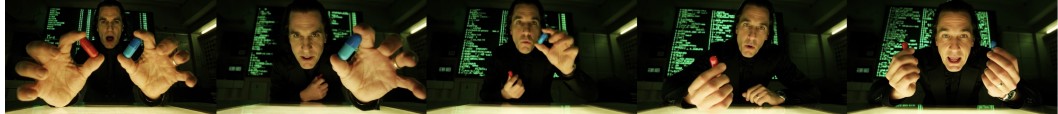

**Speech:** Swallow the blue pill and this chapter closes for good. You'll wake to the morning light and all the things you believe will stay real. Or choose the red pill and this illusion won't hold you anymore. I'll show you how many unpatched cracks lie beneath this world that feels so solid.

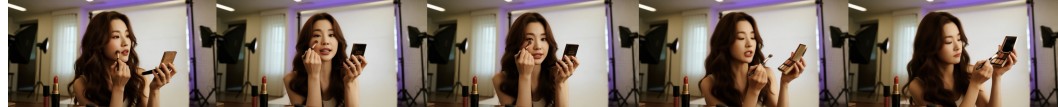

**Speech:** A blue eyeliner was applied to the inner section of the subject's lower lash line, while a purple eyeliner was applied to the outer section. For blush, a specific shade from the eyeshadow palette was utilized.

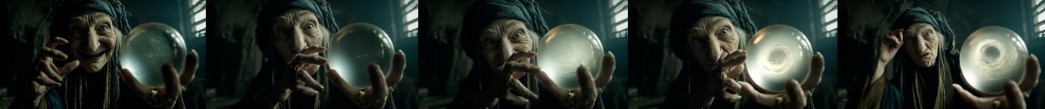

**Speech:** Crystal ball glows, wrinkled hands tell secrets. Futures twist and turn. Laugh as they beg for mercy.

Figure 6: **Reasoning Process Visualization.** The generated motion matches the speech content and the character's style in the image. We also provide the corresponding reasoning process.

**Evaluation Metrics.** We employed a multi-faceted protocol with objective, subjective, and MLLM-based metrics for a comprehensive assessment.

- **Objective Metrics:** We measured generation quality via Fréchet Inception Distance (FID) (Heusel et al., 2017) and Fréchet Video Distance (FVD) (Unterthiner et al., 2019), alongside no-reference Image Quality (IQA) and Aesthetics (ASE) scores from Q-align (Wu et al., 2023). We assessed audio-visual synchronization and hand quality using several metrics. For audio-visual synchronization, we primarily employed the widely used Sync-Confidence (Sync-C) metric from SyncNet (Chung & Zisserman, 2017). However, recognizing its potential limitations in multi-person scenarios, where it can yield misleadingly high scores for non-speaking faces, we additionally adopted the Sync-Distance (Sync-D) metric specifically for evaluating these cases. For hand quality, we used Hand Keypoint Confidence (HKC) and Hand Keypoint Variance (HKV) as proposed by Lin et al. (2025a).

- **Subjective Metrics (Human Evaluation):** As objective metrics often fail to capture high-level semantics and perceptual realism, we conducted a comprehensive user study with 40 participants. The study consisted of two main protocols.

  The first was a **pairwise comparison**, where participants viewed randomized pairs of videos from different methods. This comparison involved two distinct evaluation tasks:

  - Overall Quality ("Which is better?"): Participants chose the video with the best overall quality. From these preferences, we computed a Good/Same/Bad (GSB) score, defined as $(\text{Wins} - \text{Loses})/(\text{Wins} + \text{Loses} + \text{Ties})$.
  - Artifact Analysis ("Which is less bad?"): Participants identified specific flaws in the less preferred video, including Lip-sync Inconsistency (LSI), Motion Unnaturalness (MU), and Image Distortion (ID), to calculate a defect rate for each category.

  The second protocol was a **best-choice selection**, where participants selected the single best video from a group of all competing methods, yielding a Top-1 selection rate as a direct measure of overall appeal.

- **MLLM-based Evaluation:** To complement our human studies with a scalable and reproducible method for assessing semantic quality, we also employed an MLLM-based evaluation protocol. This automated approach assesses high-level aspects such as contextual coherence and logical consistency, which are difficult for traditional metrics to capture. The detailed instructional prompt used for this evaluation is provided in Appendix D.2.

# D  FURTHER ANALYSIS OF THE AGENTIC SYSTEM

## D.1  REASONING PROCESS DEMONSTRATION

In this section, we illustrate the workflow of our MLLM-based agentic system. The process begins with an **Analyzer** module, which uses a guided, step-by-step probing method to perform a multi-dimensional analysis of the multimodal input, including linguistic content, emotional state, and behavioral intent. Based on the analysis results, a **Planner** module then infers a plausible and logically consistent sequence of actions for the character.

Figure 6 provides several visual examples generated by this process. Below, we detail the corresponding reasoning steps the system took to generate the final video. The examples below illustrate how the system deconstructs complex inputs and formulates a detailed action plan, or "schedule", to guide the video synthesis. This process yields a schedule that is not only rational but also guides the synthesis model to produce videos that are dynamically rich and logically coherent.

It is important to emphasize that our core contribution is the modular and framework-level design itself. By introducing a dual-system design, we endow the avatar model with "an active mind" and context-awareness, thereby achieving a level of vividness that is difficult for existing models to replicate. This provides a promising baseline for the community. We envision that this agentic system can be significantly enhanced in several ways. Future work could involve applying advanced training techniques to bolster reasoning, integrating more powerful MLLMs for planning, or even pursuing end-to-end training of the core LLM. We believe these directions are all worthy of further exploration to raise the performance ceiling of our framework.

---

**Illustrating the Reasoning Process for the Sample in Row 1**

**Analysis Process**

> **Language Recognition: Identify the language used in the audio, such as English, Chinese, Korean, Japanese, Thai, etc.**
> The language used in the audio was identified as English.

> **Speech Transcription: Please transcribe the person's speech input completely and accurately. Do not omit or misspell any words while preserving the original expressions and sentence structure.**
> "Swallow the blue pill, and this chapter closes for good. You'll wake to the morning light, and all the things you believe will stay real. Or choose the red pill, and this illusion won't hold you anymore. I'll show you how many unpatched cracks lie beneath this world that feels so solid."

> **Type Identification: Determine the type of audio - just answer (speech) or (singing) only.**
> The audio is classified as speech, not singing.

> **Emotion Analysis: Briefly analyze the emotional state of the speaker or performer.**
> The speaker's tone is serious and contemplative, reflecting a significant decision or revelation.

> **Intention and Purpose Analysis: Analyze the intention and purpose of the speaker or performer, and explain the information or motivation they aim to convey.**
> The speaker's intention is to present a critical choice between blissful ignorance (blue pill) and the pursuit of truth (red pill).

> **Content Analysis: Divide the audio into multiple segments for separate analysis, and analyze and summarize the content expressed in the speech, with each segment lasting seconds.**

---

The audio was segmented into four logical parts, each approximately 4.8 seconds in duration.

**Multi-shot Analysis: Given the multi-shot Content Analysis for audio segments and above all information, analyze the intention and purpose of each segment.** Segment 1 introduces the blue pill, Segment 2 details its consequence, Segment 3 introduces the red pill, and Segment 4 details its consequence.

---

**Analysis Summary**

**Language:** English

**Transcription:** Swallow the blue pill, and this chapter closes for good. You'll wake to the morning light, and all the things you believe will stay real. Or choose the red pill, and this illusion won't hold you anymore. I'll show you how many unpatched cracks lie beneath this world that feels so solid.

**Audio Type:** speech

**Emotion:** The speaker's tone is serious and contemplative, reflecting a significant decision or revelation.

**Overall Intention:** The speaker's intention is to present a critical choice between blissful ignorance (blue pill) and the pursuit of truth (red pill).

**Segmentation Method:** The audio was segmented into four logical parts, each approximately 4.8 seconds in duration.

**Segment Intentions:**
- **Segment 1:** Introduces the blue pill.
- **Segment 2:** Details its consequence.
- **Segment 3:** Introduces the red pill.
- **Segment 4:** Details its consequence.

---

**Generated Schedule**

**Segment 1:** A man leans forward, retracting his left hand holding a red pill while aggressively pushing his right hand, which holds a blue pill, directly toward the camera. His eyes are wide and intense, he speaks with a serious expression, and his upper body tilts forward with the motion.

**Segment 2:** The man holds his pose, keeping his right hand with the blue pill prominently extended forward. He continues to speak, his intense and persuasive gaze fixed on the camera. He makes slight, subtle shifts in his posture but the focus remains on his face and the offered pill.

**Segment 3:** The man swiftly retracts his right hand holding the blue pill and thrusts his left hand, now holding the red pill, forward toward the camera. He leans in again, his expression shifting to a dramatic, wide-eyed challenge as he presents the alternative choice, his mouth forming the words.

**Segment 4:** While keeping the red pill extended in his left hand, the man brings his right hand with the blue pill forward again, presenting both choices simultaneously to the camera. He leans in even closer, his eyes widening in a manic, intense stare as he speaks, emphasizing the gravity of the decision.

---

**Illustrating the Reasoning Process for the Sample in Row 2**

**Analysis Process**

**Language Recognition: Identify the language used in the audio, such as English, Chinese, Korean, Japanese, Thai, etc.**
The language used in the audio was identified as Chinese.

**Speech Transcription: Please transcribe the person's speech input completely and accurately. Do not omit or misspell any words while preserving the original expressions and sentence structure.**
The content of the speech is "For the lower eyeliner, draw the front part with blue eyeliner, the back part with purple eyeliner, and for the red, use this one from the eyeshadow palette." (English translation)

**Type Identification: Determine the type of audio - just answer (speech) or (singing) only.**
The audio is classified as speech, not singing.

**Emotion Analysis: Briefly analyze the emotional state of the speaker or performer.**
The speaker's tone is neutral and informative, consistent with providing instructions.

**Intention and Purpose Analysis: Analyze the intention and purpose of the speaker or performer, and explain the information or motivation they aim to convey.**
The speaker's intention is to provide instructions on how to apply eye makeup, specifically guiding the listener on using blue, purple, and red shades for a particular look.

**Content Analysis: Divide the audio into multiple segments for separate analysis, and analyze and summarize the content expressed in the speech, with each segment lasting seconds.**
The audio was segmented into two logical parts, with the first lasting 6 seconds.

**Multi-shot Analysis: Given the multi-shot Content Analysis for audio segments and above all information, analyze the intention and purpose of each segment.**
Segment 1 instructs on applying blue and purple eyeliner to the lower lash line, while Segment 2 suggests using a specific red eyeshadow to complete the look.

---

**Analysis Summary**

**Language:** Chinese

**Transcription:** For the lower eyeliner, draw the front part with blue eyeliner, the back part with purple eyeliner, and for the red, use this one from the eyeshadow palette.

**Audio Type:** speech

**Emotion:** Neutral and informative.

**Overall Intention:** To provide instructions on how to apply eye makeup using blue, purple, and red shades.

**Segmentation Method:** The audio was segmented into two logical parts.

**Segment Intentions:**
- **Segment 1:** Instructs on applying blue and purple eyeliner.

- **Segment 2:** Suggests using a red eyeshadow to complete the look.

---

**Generated Schedule**

**Segment 1:** A beautiful East Asian woman holds a small compact mirror and meticulously applies eyeshadow to her upper eyelid using a fine-tipped makeup brush, her expression focused as she looks into the mirror.

**Segment 2:** She then looks directly at the camera while speaking, and gestures with the brush towards the open eyeshadow palette in her other hand, as if explaining the next step in a makeup tutorial.

---

**Illustrating the Reasoning Process for the Sample in Row 3**

**Analysis Process**

**Language Recognition: Identify the language used in the audio, such as English, Chinese, Korean, Japanese, Thai, etc.**
The language used in the audio was identified as English.

**Speech Transcription: Please transcribe the person's speech input completely and accurately. Do not omit or misspell any words while preserving the original expressions and sentence structure.**
The complete transcription of the speech is: "Crystal ball glows, wrinkled hands tell secrets, futures twist and turn laugh as they beg for mercy."

**Type Identification: Determine the type of audio - just answer (speech) or (singing) only.**
The audio is classified as speech, not singing.

**Emotion Analysis: Briefly analyze the emotional state of the speaker or performer.**
The speaker's tone is serious and contemplative, suggesting a mystical or prophetic mood.

**Intention and Purpose Analysis: Analyze the intention and purpose of the speaker or performer, and explain the information or motivation they aim to convey.**
The speaker's intention is to evoke a sense of mysticism and foresight, describing a scene of fortune-telling or prophecy where futures are revealed.

**Content Analysis: Divide the audio into multiple segments for separate analysis, and analyze and summarize the content expressed in the speech, with each segment lasting seconds.**
The audio was segmented into two logical parts, with the first lasting 4.8 seconds.

**Multi-shot Analysis: Given the multi-shot Content Analysis for audio segments and above all information, analyze the intention and purpose of each segment.**
Segment 1 sets a mystical scene with a glowing "crystal ball", while Segment 2 adds a human element of ancient wisdom with "wrinkled hands" revealing secrets.

---

**Analysis Summary**

> **Language:** English
>
> **Transcription:** Crystal ball glows, wrinkled hands tell secrets, futures twist and turn laugh as they beg for mercy.
>
> **Audio Type:** speech
>
> **Emotion:** Serious and contemplative, with a mystical or prophetic mood.
>
> **Overall Intention:** To evoke a sense of mysticism and describe a scene of fortune-telling.
>
> **Segmentation Method:** The audio was segmented into two logical parts.
>
> **Segment Intentions:**
> - **Segment 1:** Sets a mystical scene with a glowing crystal ball.
> - **Segment 2:** Adds a human element of wisdom with wrinkled hands revealing secrets.
>
> ---
>
> **Generated Schedule**
>
> **Segment 1:** The woman leans forward slightly, her gaze fixed and intense, her gnarled right hand gesturing mysteriously near her face. The crystal ball in her left hand begins to emit a faint, ethereal glow, hinting at the power stirring within.
>
> **Segment 2:** A swirling vortex of white light suddenly manifests and spins within the crystal ball. The witch's expression shifts dramatically; her eyes widen and her mouth opens in a gasp of shock or revelation as she witnesses the unfolding future.

We also provide the full system prompt for the Planner MLLM in the supplementary material, as it is too verbose to include even in the appendix.

## D.2 FURTHER QUANTITATIVE EVALUATION

To further validate our design, we conducted supplementary MLLM-based evaluations using Gemini-2.5-Pro (Comanici et al., 2025), a leading MLLM with strong audio-visual comprehension. This evaluation focused on three key dimensions: Overall Quality, Prompt Following, and Motion-Context Coherence. As shown in Table 5, the results indicate that our full model substantially outperforms an ablation using OmniHuman-1's conditioning method (reference attention and audio cross-attention). Furthermore, our approach demonstrates a significant advantage when compared to a variant of our model lacking the agentic reasoning module. These considerable improvements underscore the effectiveness of our core designs. In particular, the notable enhancement in Motion-Context Coherence highlights our model's superior ability to generate action sequences with logical and self-coherence. The table also includes MLLM-based pairwise comparisons for our other architectural ablations, confirming that each proposed module contributes positively to the final performance.

Finally, we provide an additional subjective user study comparing our method against the recent open-source model, Wan2.2-S2V. As shown in Table 6, our model achieves significantly better performance, both in the overall GSB preference score and across all measured artifact dimensions.

**MLLM Evaluation Prompt and Protocol.** We provide the detailed instruction prompt used for the MLLM-based pairwise comparison. During the evaluation, videos from our method and the competing method were stitched side-by-side into a single video file to be fed into the MLLM. To mitigate positional bias, the placement of the two videos (left or right) was randomized for each comparison, and each sample pair was evaluated five times to ensure robust results.

Table 5: **MLLM-based Evaluation**. Pairwise comparison results (GSB Score) with individual scores for Overall Quality, Prompt Following, and Motion-Context Coherence.

| Baseline | Proposed | Overall Quality | Prompt Following | Motion-Context Coherence |
|---|---|---|---|---|
| Ours w/ OmniHuman-1 Conditioning | **Ours (Full Model)** | 15% | 14% | 24% |
| Ours w/o Agentic Reasoning | **Ours (Full Model)** | 26% | 29% | 28% |
| Ours w/ Ref. Attention | **Ours (Full Model)** | 10% | 7% | 8% |
| Ours w/o MM Warm-up | **Ours (Full Model)** | 10% | 8% | 9% |
| *Comparison between reasoning module variants* | | | | |
| Ours w/ Single-Step Reasoning | **Ours (Full, Multi-Step)** | 13% | 16% | 7% |
| Ours w/o Agentic Reasoning | Ours w/ Single-Step Reasoning | 22% | 24% | 17% |

Table 6: **Subjective Evaluation vs. Wan2.2-S2V.** Subjective evaluation against Wan2.2-S2V, including a GSB preference score (Which is better?) and a pairwise comparison of artifacts (Which is less bad?).

| Method | LSI↓ | MU↓ | ID↓ | GSB↑ |
|---|---|---|---|---|
| Wan2.2-S2V | 0.24 | 0.60 | 0.21 | −0.71 |
| Ours (Full Model) | 0.04 | 0.08 | 0.02 | +0.71 |

---

**Prompt for MLLM-based Pairwise Video Evaluation**

**Task: Multi-dimensional Comprehensive Evaluation of AI-Generated Videos**

1. **Role** You are a comprehensive review expert specializing in filmmaking, visual storytelling, and multimedia arts. Your mission is to conduct an objective, in-depth, and detailed comparative evaluation of two AI-generated videos based on a rigorous and professional set of multi-dimensional criteria.
   **Core Requirements:**
   - **Objectivity:** Disregard personal preferences and base your judgment solely on the provided criteria and input information.
   - **Professionalism:** Conduct your analysis from a professional perspective.
   - **Focus Area:** Your evaluation should primarily focus on the actors and characters in the video, including their expressions and movements. You may moderately disregard other non-primary elements such as backgrounds and props.

2. **Core Evaluation Dimensions** You will comparatively evaluate Video 1 (left) and Video 2 (right) across the following three core dimensions.

   **Dimension 1: Global Quality** - A comprehensive assessment of the video's overall artistic appeal, technical execution, and viewer experience.

   **Evaluation Points:**
   - **Image Quality:** Clarity, color representation, lighting, and the presence of any noticeable noise or artifacts.
   - **Motion Naturalness:** Whether movements are fluid, conform to the laws of physics, and are free from stiffness or the uncanny valley effect.
   - **Expressiveness:** Whether facial expressions are vivid, nuanced, and capable of accurately conveying emotions.
   - **Aesthetics & "Wow" Factor:** Does the video possess a unique visual style or artistic beauty? Does it offer surprises in creativity, composition, or visual effects?
   - **Viral Potential:** Which video is more likely to resonate with an audience and inspire sharing?

   **Dimension 2: Action Control & Prompt Adherence** - An assessment of how accurately the video responds to specific "action" and "expression" instructions in the user's prompt.

**Evaluation Points:**

- ⊖ **Instruction Fidelity:** Are the core actions and expressions executed strictly as described in the prompt? Are there any omissions or misinterpretations?
- ○ **Action Completion:** Are the instructed actions fully and clearly rendered? Is the sequence of movements coherent? Is the sense of power, rhythm, and amplitude appropriate?
- ○ **Detail Representation:** How well are complex or subtle actions (e.g., hand gestures, eye contact) reproduced?

**Dimension 3: Overall Content Coherence & Logicality** - An assessment of the logical, contextual, and emotional consistency between the video content and the audio content.

**Evaluation Points:**

- ⊖ **Contextual Appropriateness:** Considering the audio, are the character's actions and expressions logical and fitting for the scene and intent?
- ○ **Audiovisual Synchronization:** Does the rhythm of the on-screen actions match the audio? Are the movements human-like, natural, and reasonable?
- ○ **Narrative Unity:** Do the visual and audio elements work together to serve a unified theme or emotional tone? Do they enhance each other, or do they conflict?

---

### 3. Evaluation Task & Output Format

**Input Information:**

1) **Video for Evaluation:** A side-by-side video file with unified audio.
   - Video 1: Located on the left side of the screen.
   - Video 2: Located on the right side of the screen.
2) **Original Prompt:** The text instruction used to generate both videos.

**Task Flow:**

1) Carefully analyze the requirements.
2) Simultaneously watch and compare Video 1 and Video 2.
3) Conduct a point-by-point comparative analysis based on the Core Evaluation Dimensions.
4) Strictly adhere to the JSON format below to output your final evaluation.

**Output Requirements:**

Based on your comparative analysis, provide a final score for each dimension using the following rubric.

- `-1`: The left video is significantly better.
- `0`: The two videos are comparable, or each has its own merits.
- `1`: The right video is significantly better.

**JSON Output Format:**

```
{
  "global_quality": {
    "score": "Enter -1, 0, or 1"
  },
  "action_control": {
```

```
      "score": "Enter -1, 0, or 1"
    },
    "overall_content_coherence": {
      "score": "Enter -1, 0, or 1"
    }
}
```

## E  ADDITIONAL QUALITATIVE RESULTS AND VISUAL ABLATIONS

In this section, we provide additional visual comparisons and ablation studies to further demonstrate the effectiveness of our method.

**Qualitative Comparison with Baselines.** Figure 7 showcases a comparative analysis of our method against recent approaches. The first example highlights a common limitation in existing methods: a tendency to produce static or minimally varied outputs. Even OmniHuman-1, despite generating more varied gestures, fails to establish a meaningful correlation with the audio and struggles with temporal consistency, causing objects like the held capsule to flicker. Furthermore, its output often appears over-sharpened, which can degrade visual quality, particularly with high-saturation inputs. This issue of temporal instability is exacerbated in longer generation sequences, as shown in the second example. There, competing methods either remain static or introduce significant visual artifacts, such as discoloration, when motion is attempted. Our method, in stark contrast, maintains both high-dynamic motion and excellent visual stability throughout the extended duration. This dual achievement is a direct result of our design: the logical, speech-matched motion stems from our agentic system, while the high-quality, consistent object interaction is enabled by our conditioning method's powerful fusion of multimodal cues.

In Figure 12, we further present a visual comparison with the state-of-the-art method, OmniHuman-1. The corresponding speech content for each video is provided. As can be seen, our method demonstrates significantly stronger semantic relevance and logical correlation between the audio and the generated actions. For instance, in the first video, the character turns her head when calling out "Mary"; in the second, she performs the specific actions of applying eyeliner and gesturing towards the eyeshadow palette as described; and in the third, the crystal ball glows and changes in response to the wizard's incantation. These high-level, context-aware results are difficult to capture with objective metrics and remain a challenge for existing methods. We encourage readers to view the examples on our project page for a more intuitive understanding of the correlation between generated motion, speech content, and character intent.

**Visual Ablation of the Pseudo Last Frame (PLF).** In Figure 8, we demonstrate the critical role of our proposed Pseudo Last Frame (PLF). We compare generated results with and without the PLF module under conditions of significant camera and character motion. The difference is stark: without the PLF, the generated sequence undergoes drastic changes in character identity and scene content, rendering it unusable. In contrast, the version equipped with the PLF successfully maintains high-quality visuals and character consistency while accommodating high-dynamic movements and large camera shifts.

Furthermore, Figure 9 provides a deeper analysis of the PLF's mechanism by comparing it against two alternatives: a standard reference attention mechanism and the PLF with varying RoPE distance offsets. While reference attention can respond to the text prompt, it gradually reverts to the reference image during autoregression, exhibiting a "reset" behavior. The PLF, however, effectively mitigates this issue. By increasing the RoPE distance offset, we can strike a better trade-off, enabling the model to preserve content from the reference image while still generating significant motion. For instance, with a 'rope-30' offset, the first example shows a smooth transition from a side profile to a full-frontal view of the character. A similar effect is observed in the second example, where the 'rope-30' offset locks the camera's viewpoint at the desired state, preventing a reset.

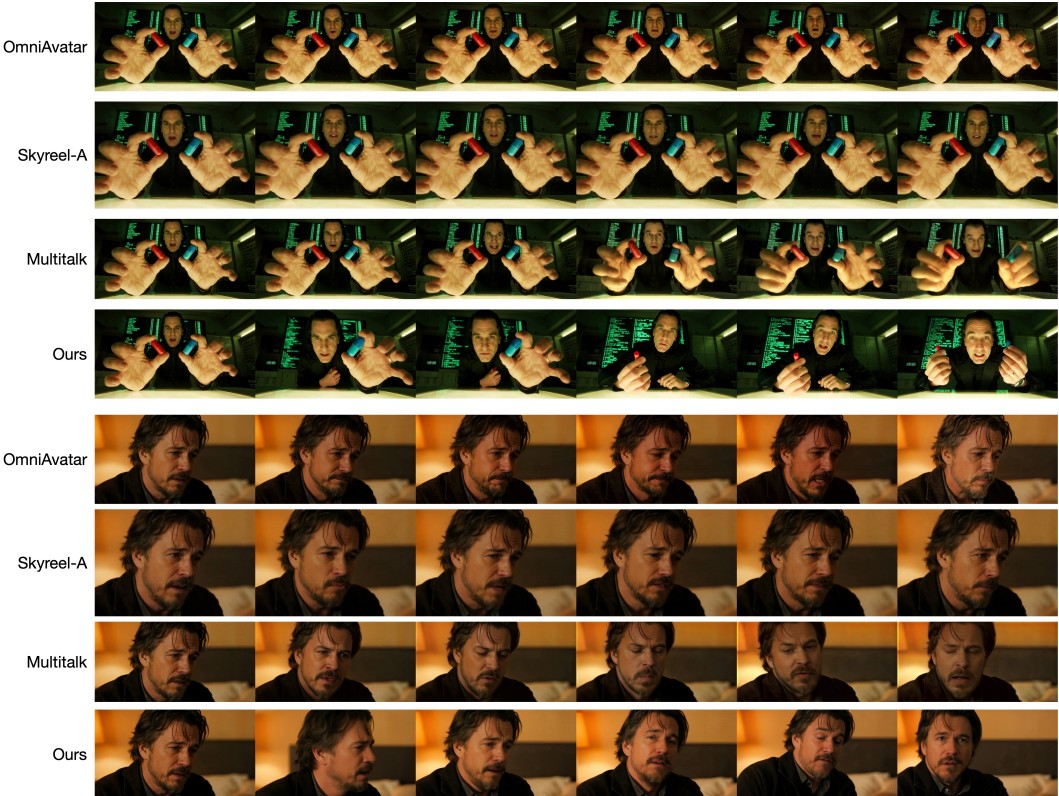

Figure 7: **Qualitative Comparison with Recent Methods.**

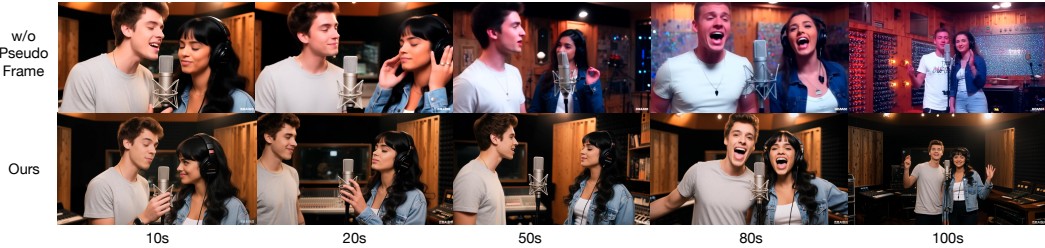

Figure 8: **Effectiveness of PLF in Preserving Content Consistency.**

In summary, the visual results in Figures 8 and 9 collectively demonstrate that the PLF module is essential for: (1) enabling large, continuous motions without the "reset" problem, and (2) simultaneously preserving the stability and consistency of image quality, color, and character identity.

**Visual Ablation of the MM-Warmup.** Figure 10 presents an ablation study on the MM-Warmup stage. Without this stage, the model exhibits two primary deficiencies. First, the resulting motion appears less dynamic, contributing to a subtle reduction in expressiveness. More critically, the quality of hand gesture synthesis deteriorates significantly, with an increase in artifacts on local body parts, indicating a compromised ability for holistic synthesis. This comparison underscores the efficacy of our approach: by integrating the proposed MM-Warmup with our MM-DiT architecture, we successfully adapt a text-and-image-to-video (T2V) model into a high-fidelity video avatar model, maintaining exceptional multimodal modeling capabilities and overall synthesis quality.

**Visual Ablation of the Reflection Process.** We visualize the impact of our optional reflection process in Figure 11. Without reflection (top row), the model generates an action schedule in a single pass, which can lead to logical inconsistencies. For instance, after the action "Takes out letter," the model generates "Rubs the surface," causing the letter to vanish and breaking semantic continuity. In

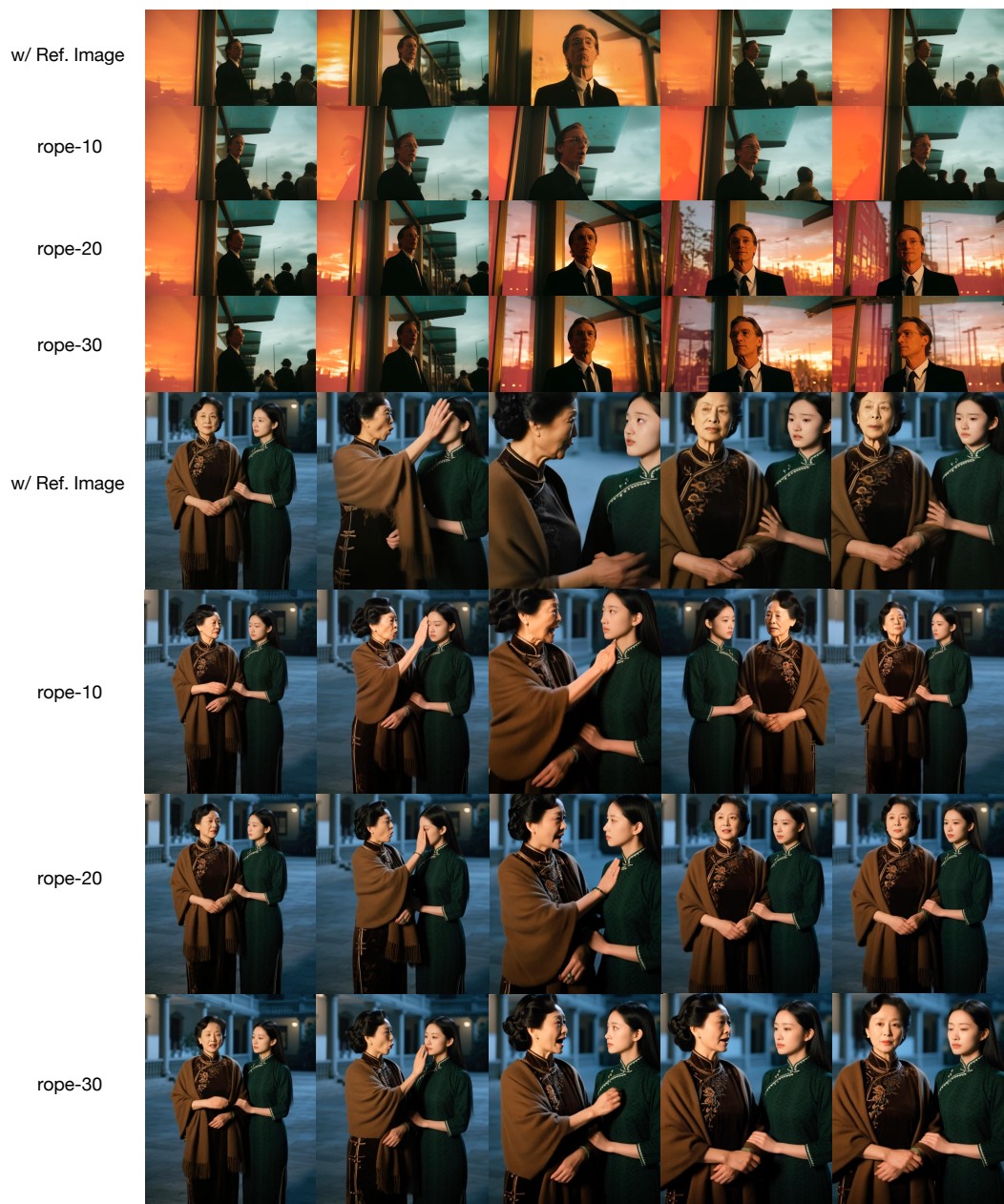

Figure 9: **Effectiveness of PLF in Maintaining Motion Dynamics.**

contrast, our model with reflection (bottom row) revises the plan after generating the first segment. It observes the outcome of "Takes out letter" and corrects subsequent actions to be relevant to the theme of letter-reading, ensuring logical progression and mitigating error accumulation. However, as this reflection process introduces additional inference overhead, it was disabled for the quantitative comparisons in this paper. We also investigated injecting reasoning latents directly into the synthesis model. While this encouraged more nuanced facial expressions, it also suppressed large, dynamic actions. As this appeared to be an aesthetic trade-off rather than a clear improvement, we excluded it from our final model configuration.

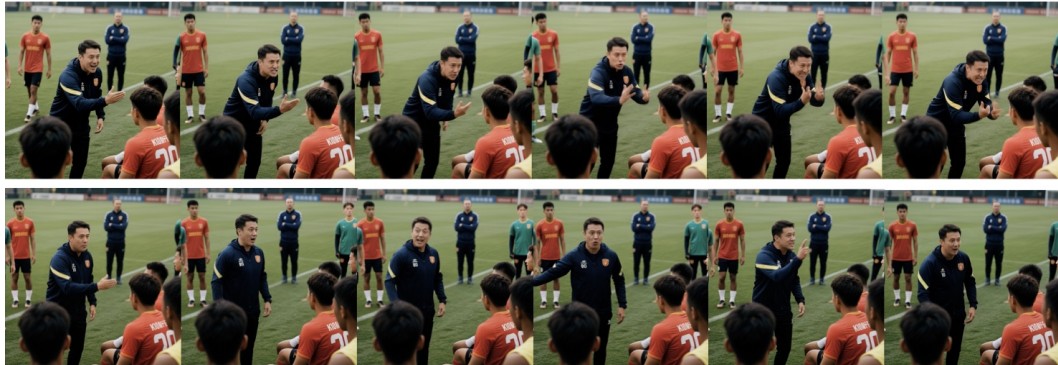

Figure 10: **Qualitative results of the reflection process.** Without reflection (first row), an ill-planned action ("Rubs the surface") causes object inconsistency. With reflection (second row), the model revises its plan to a more logical action, ensuring consistency.

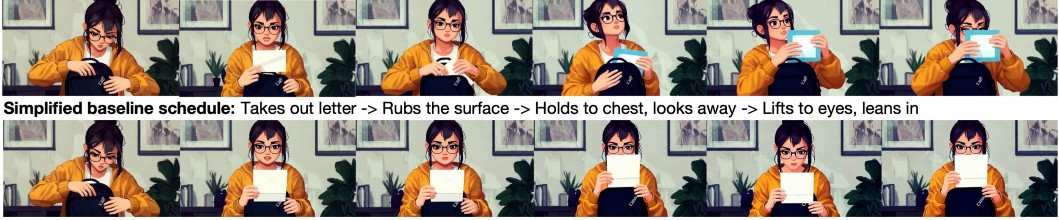

Figure 11: **Qualitative results of the reflection process.** Without reflection (first row), an ill-planned action ("Rubs the surface") causes object inconsistency. With reflection (second row), the model revises its plan to a more logical action, ensuring consistency.

## F    LATENCY OF THE AGENTIC SYSTEM

The introduction of our agentic reasoning module adds a fixed, upfront latency of approximately 20-30 seconds for its "thinking" process, a measurement taken without any engineering optimizations. This cost is constant regardless of video length, and we argue it is a justifiable trade-off for two primary reasons. **First,** this latency is minor when contextualized against the 5-6 minutes that mainstream Diffusion Transformer (DiT) based video generation models require for a 10-second clip. We also anticipate this cost will diminish significantly with the rapid advancement of LLM inference technologies. **Secondly,** this reasoning phase is precisely what enables the substantial improvements in expressive power and contextual coherence that characterize our method. This process mirrors the "slow thinking" of deliberate human cognition, making it a necessary step for achieving higher-level intelligence, not merely a computational overhead. Complex reasoning is inherently time-consuming. Therefore, we consider this latency a reasonable and even desirable price for the significant leap in generation quality.

## G    STATEMENTS

**Ethics Consideration.** Our core contribution is a novel paradigm for video avatar generation. By simulating a dual-system cognitive framework, our model achieves a new level of expressive capability and logical coherence, moving beyond the limitations of single-process generation. While this advancement opens exciting possibilities for creative applications like AI-driven film production, we recognize the potential for misuse associated with highly realistic avatar technologies.

To address these ethical concerns, we advocate for a robust framework of responsible deployment. Given its potentially powerful capabilities, this model requires strict access control to prevent potential abuse. Although current results may still bear subtle artifacts that can serve as an informal deterrent, proactive safeguards are essential. We strongly recommend the following measures: (1)

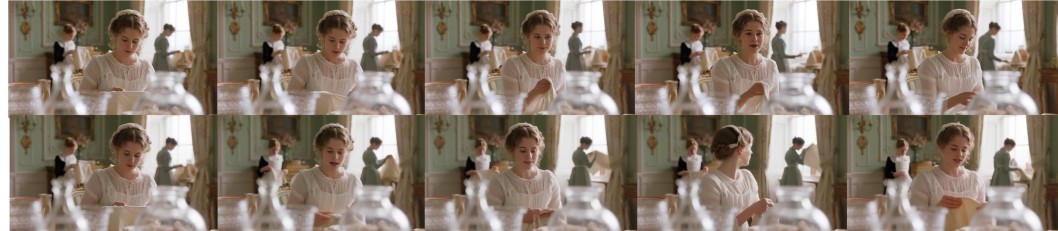

**Speech:** Mary, pass me the ivory thread. This hem, it needs to be perfect

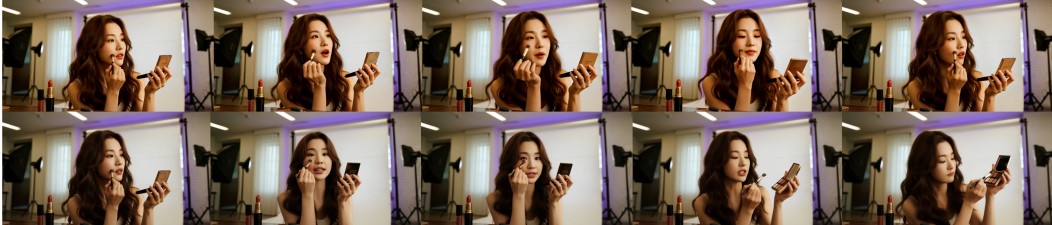

**Speech:** A blue eyeliner was applied to the inner section of the subject's lower lash line, while a purple eyeliner was applied to the outer section. For blush, a specific shade from the eyeshadow palette was utilized.

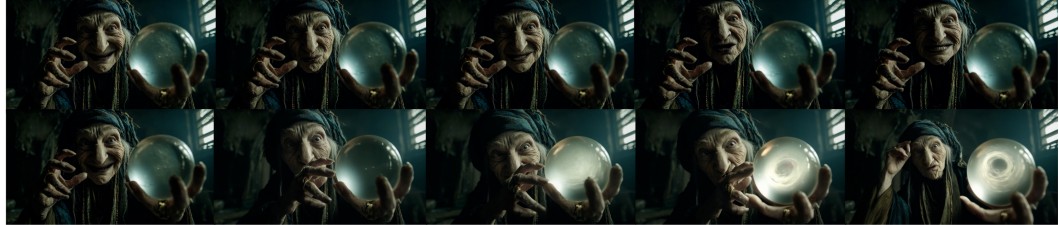

**Speech:** Crystal ball glows, wrinkled hands tell secrets. Futures twist and turn. Laugh as they beg for mercy.

Figure 12: **Qualitative comparison with OmniHuman-1 (Lin et al., 2025b), a state-of-the-art model representing the predominant conditioning scheme of audio cross-attention and image reference attention.** For each pair of examples, our model (bottom row) generates actions with higher semantic consistency to the speech prompt than the baseline (top row). For example, our model correctly depicts a character applying makeup and a glowing crystal ball as described in the speech, actions which are absent in the baseline's results.

applying prominent, visible watermarks to all generated content to clearly label it as AI-generated; (2) implementing filtering algorithms to reject inappropriate or malicious input prompts and to review output content; and (3) embedding traceable, invisible watermarks to ensure accountability and aid in source identification if misuse occurs. By integrating these safety protocols, we can help ensure that our technology fosters creativity while minimizing the risks of malicious applications such as fraud or disinformation.

**LLM Usage Statement.** We utilized Large Language Models (LLMs) as a general-purpose assistive tool in the preparation of this manuscript. The usage was confined to two main areas: (1) refining language and word choice for pre-written sentences by seeking explanations to determine the most effective phrasing, and (2) assisting with technical aspects of document preparation, such as generating syntax for LaTeX tables and debugging compilation errors. The authors are fully responsible for all intellectual content of the paper.

**Reproducibility Statement.** To ensure the full reproducibility of our work, we provide comprehensive details regarding our methodology, data, implementation, and evaluation procedures across the main paper and its appendices. Specifically, we detail: (1) the core architecture of our agentic system and the MMDiT model in Section 3; (2) complete implementation details, including all hyperparameters and the multi-stage training procedure, in Appendix B; (3) our data curation pipeline, with specific filtering tools (e.g., PySceneDetect, Q-Align) and a statistical analysis of the dataset, also in Appendix B; and (4) our agentic system and evaluation framework, which specifies the MLLM models used (miniCPM-o and Seed-1.5-VL), instruction prompts, and the complete set of objective and subjective evaluation metrics in Appendices D and C. Finally, while we provide

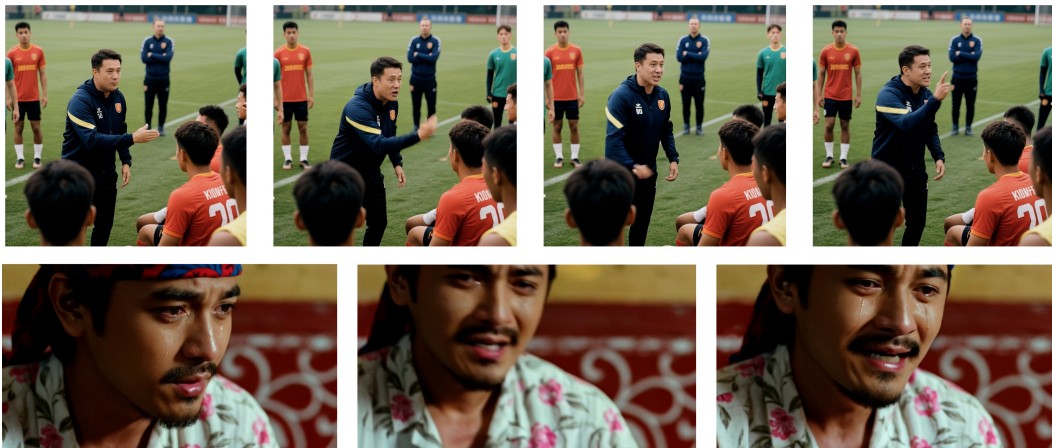

Figure 13: **Limitations of our method.** Despite its strong performance, our method exhibits limitations in both synthesis and reasoning. On the synthesis level, the model can struggle with fine details during rapid movements, leading to hand artifacts (row 1) and degraded facial identity during large head turns (row 2). Separately, on the reasoning level, the agent can generate motions that, while plausible, are occasionally over-articulated (e.g., excessive gesturing), lacking the subtlety required for cinematic performance.

these details to ensure technical reproducibility, we reiterate our ethical recommendation that for responsible deployment, access to the trained models should be managed in a controlled environment to monitor usage and prevent potential misuse.

## H    LIMITATIONS

**Limitations and Future Work.**    While our method achieves strong overall performance, it is not without its limitations, as visualized in Figure 13. These challenges primarily fall into two categories: minor artifacts at the synthesis level and occasional imperfections in motion reasoning. To address these issues and further advance our model's capabilities, we have identified two key directions for future work. First, to enhance the model's fundamental generation quality and mitigate synthesis artifacts, we plan to continue scaling up by training on larger, higher-quality datasets. Second, to improve the alignment between high-level reasoning and low-level motion synthesis, we will investigate more sophisticated integration strategies, with a particular focus on the end-to-end joint training of the DiT and the LLM.

