# OpenReview forum: "Instilling an Active Mind in Avatars via Cognitive Simulation"
_ICLR.cc/2026/Conference — ICLR 2026 Oral_

### Official Review · Reviewer_fLxy · 2025-10-19

**Soundness:** 4
**Presentation:** 4
**Contribution:** 3
**Rating:** 8
**Confidence:** 5

**Summary:**

This study proposes an avatar video generation framework consisting of two synergistic systems: System 1 is responsible for audio-to-video generation, and System 2 undertakes tasks of context awareness and logical inference. Through the collaboration of the two systems, the framework can better understand contextual semantic information to generate avatar videos; verified by extensive experiments, this method can significantly improve the quality and vividness of the generated content.​

**Strengths:**

- The proposed dual-system avatar generation framework makes full use of the capabilities of MLLMs and optimizes the performance of video generation models. It provides an accurate analysis of the current dilemmas in avatar generation tasks, demonstrating strong innovation and inspiration in this field. Moreover, the dual-system model is expected to be applied to more general video generation frameworks, reflecting its high potential.​
- It puts forward optimization details such as "pseudolast-frame". Aiming at the problem of error accumulation in autoregressive generation in the field of video generation, this method effectively alleviates this issue and may significantly improve the performance in real-time streaming generation scenarios.​
- The experimental design is comprehensive and extensive, covering a large number of comparative experiments, ablation experiments, and professional user studies, which provides strong support for the research conclusions.​

**Weaknesses:**

In the current research, the combination of MLLM and MMDiT lacks tightness, and MLLM tends to play the role of a "prompt refiner".

**Questions:**

The following points need to be further clarified:​
- What are the core differences between the application of MLLM in this research and existing prompt refiner methods?​
- What specific technical efforts has the authors made to enhance the synergy between MLLM and MMDiT?​

---

> ### Author Response · Authors · 2025-11-19
>
> We are very grateful to the reviewer for acknowledging the core designs of our method, the dual-system framework and the Pseudo Last Frame (PLF), as well as our experimental validation. It is gratifying to receive such insightful feedback. We hope that our answers below will resolve any further concerns.
>
> > Regarding the Novelty and Synergy of the MLLM Framework
>
> The primary difference between our MLLM agent and existing prompt refiner methods lies in the scope and nature of the task. Most prompt refiners perform text-to-text expansion, enriching a simple text prompt into a more detailed one. In contrast, our MLLM-based agentic system performs a more complex task of multi-modal information mapping. It synthesizes information from multiple modalities, including the reference image, audio content, and character persona, into a comprehensive 'plan' or 'guidance' for the rendering model (the MMDiT). This process is not simple text expansion; it is a holistic combination of reasoning and planning that considers context, motion coherence, character traits, and visual style to generate the necessary instructions for the renderer.
>
> To specifically enhance the synergy between the MLLM (planner) and the MMDiT (executor), we introduced several technical contributions detailed in Section 3.2. This focus on extensibility further distinguishes our framework from simple prompt refiners:
> 1. The Reflective Re-planning Process: We designed the agentic system to be extensible, allowing for a feedback loop. This reflective process enables the system to iteratively refine its plan based on initial outputs or internal checks, creating a much tighter and more adaptive coupling between the planner and the renderer than a simple one-pass system.
> 2. Reasoning-Infused Audio Latents: We explored moving beyond discrete text guidance. By infusing the MLLM's reasoning directly into continuous audio latent features, we create a richer, more nuanced communication channel between the cognitive and rendering modules. This represents a significant step towards a more integrated synergy, replacing plain text with a continuous feature-based guidance signal.
>
> Ultimately, the core contribution of our work is the proposal of a novel avatar framework that simulates both 'System 1' (reactive) and 'System 2' (deliberative) functions. We believe this is a valuable and underexplored direction in the field, and we sincerely thank the reviewer for their recognition of its potential.

---

> > ### Comment · Reviewer_fLxy · 2025-11-20
> >
> > Thanks for the author's reply. My concern has been well addressed, and I am pleased to keep the original score.

---

> > > ### Author Response · Authors · 2025-11-28
> > >
> > > I appreciate your quick response and the positive feedback. It's great to hear that the concern is solved.

---

### Official Review · Reviewer_Rgov · 2025-10-30

**Soundness:** 3
**Presentation:** 3
**Contribution:** 3
**Rating:** 6
**Confidence:** 2

**Summary:**

This paper proposes a two-stage framework for avatar-centric video generation conditioned on audio, text, and reference images. In the first stage, a video plan is generated, while in the second stage, the actual video is synthesized based on this plan and the multimodal conditions. Notably, when incorporating the reference image, the paper introduces the Pseudo Last Frame (PLF) technique, which places the reference image at a distant, unreachable position in RoPE. This approach introduces reference information while avoiding forcing the model to regress to that exact image.

As far as I know, two-stage frameworks have been explored in various contexts, and the paper's emphasis on "cognitive simulation" appears to be more of a conceptual framing. Therefore, there are certain limitations in terms of novelty. However, considering the effort involved in adapting to this specific domain, the innovation of PLF, and the promising qualitative and quantitative results, I am tentatively inclined to accept this paper.

**Strengths:**

1. The paper is clearly written and easy to follow. The figures and tables are well-designed and visually appealing.

2. The proposed PLF is quite interesting and demonstrates effectiveness.

3. The quantitative metrics show notable improvements. The visualization and supplementary video results demonstrate significant quality enhancements.

**Weaknesses:**

1. The proposed two-system framework—using an LLM as a planner followed by another system/model as an executor—has been explored in various contexts including video generation, text-to-image generation [1], robotics [2], and audio generation [3]. While the application to avatar generation shows merit, the overall structural approach is relatively well-established in the literature.

2. The concept of "cognitive simulation" proposed in the paper could benefit from more rigorous justification. From my understanding, what is referred to as cognitive simulation appears to largely correspond to the planner-executor framework mentioned above. Given the prevalence of this framework across different domains, it would be helpful to clarify the specific advantages or unique aspects that the cognitive simulation framing brings to this particular application.

**Questions:**

1. Regarding PLF, during inference, is the reference image placed only at the "pseudo last frame" position, or is it placed at both the first frame and the "pseudo last frame" position simultaneously?

2. Is there a comparison between PLF and the prior method that "conditions the model on a reference image sampled from the training video"?

3. How does PLF perform on image-to-video benchmarks? In other words, within a broader scope, is PLF a superior method for introducing reference images in image-to-video generation?

##### References

[1] Qu L, Wu S, Fei H, et al. Layoutllm-t2i: Eliciting layout guidance from llm for text-to-image generation[C]//Proceedings of the 31st ACM International Conference on Multimedia. 2023: 643-654.

[2] Kannan S S, Venkatesh V L N, Min B C. Smart-llm: Smart multi-agent robot task planning using large language models[C]//2024 IEEE/RSJ International Conference on Intelligent Robots and Systems (IROS). IEEE, 2024: 12140-12147.

[3] Liang J, Zhang H, Liu H, et al. Wavcraft: Audio editing and generation with natural language prompts[C]. ICLR 2024 Workshop on LLM Agents, 2024.

---

> ### Author Response · Authors · 2025-11-19
>
> We are truly grateful to the reviewer for their positive feedback, especially regarding our presentation and the design of the Pseudo Last Frame (PLF). This recognition is highly encouraging, and we will do our best to address their remaining concerns, hoping that our responses will further strengthen your positive assessment of our work.
>
> > Regarding the Framework's Novelty and the "Cognitive Simulation" Concept
>
> We thank the reviewer for their insightful comments regarding our framework and the framing of "cognitive simulation."
>
> We agree that planner-executor architectures are an emerging and powerful trend across various generative domains. While the high-level structure is familiar, we argue that its application to audio-driven avatar generation is non-trivial and presents unique challenges. Our work specifically explores how to adapt this paradigm to a complex, multi-modal context that requires a deep, integrated understanding of speech content, visual cues, character persona, and long-term temporal planning. Our results clearly demonstrate that this approach yields significant, measurable improvements over existing 'System 1'-based models in the avatar space. We thank the reviewer for providing these valuable references. They will be incorporated into the related work section in the final version.
>
> Regarding the term "cognitive simulation," we use it as a functional analogy to describe our goal: to imbue the avatar with human-like planning capabilities. As noted on page 2, we believe this framing is particularly apt for avatar generation, a fundamentally human-centric task, where mimicking cognitive processes offers a clear path toward more natural and believable behavior.
>
> The specific advantages of this approach are quantitatively demonstrated in our ablation studies:
>
> - Table 1 & 2: These tables show that removing the agentic reasoning module leads to a measurable decrease in motion dynamism (HKV) and a significant increase in motion unnaturalness (MU).
> - Table 5: Our full model achieves a substantial win rate (+28% GSB) in Motion-Context Coherence when compared to a baseline without the reasoning module.
>
> These results provide rigorous justification that our "cognitive simulation" framework delivers specific, measurable advantages in motion vividness and contextual appropriateness, key metrics for avatar quality.
>
> > Regarding the Pseudo Last Frame (PLF) Mechanism
>
> We thank the reviewer for these specific questions about the PLF mechanism.
>
> 1. PLF Inference Process:
>
> For the initial video segment, which does not require auto-regressive continuation, the reference image is used for both the first frame and the "pseudo last frame." For all subsequent segments, the first frame is the final frame of the preceding segment to ensure temporal continuity, while the reference image continues to be used for the "pseudo last frame." This process is described on L202 of the manuscript.
>
> 2. Comparison with Prior Methods:
>
> Yes, we do provide this comparison. The baseline method, which conditions the model on a reference image sampled from the training video (via standard reference attention), is represented by the "ref attention" condition in our experiments. The performance comparison can be found in Table 5 (comparing our full model to the "omnihuman-1 condition" and "ref attention condition") and in the ablation studies in the lower half of Table 1.
>
> 3. Performance on General Image-to-Video (I2V) Benchmarks:
>
> This is an excellent question about the broader applicability of PLF. There is a conceptual difference between our task and standard image-to-video (I2V) generation. PLF is designed with the assumption that the reference image acts as a persistent anchor for the character's identity throughout the video. In contrast, typical I2V tasks treat the input image as a starting point for more divergent and unconstrained generation.
>
> Despite this difference, PLF is a generalizable technique that can be adapted for I2V tasks in certain scenarios, particularly for injecting a consistent reference character into a generated video. **We have included examples in our updated supplementary materials (7_rebuttal folder) to demonstrate this potential application.**

---

> ### Comment · Reviewer_Rgov · 2025-11-28
>
> Thanks for the author's reply. My concerns have been largely addressed, and I am pleased to revise my score upward accordingly.

---

> > ### Comment · Reviewer_Rgov · 2025-11-28
> >
> > It appears that the score cannot be modified. Therefore, I will temporarily maintain the original score.

---

> > > ### Author Response · Authors · 2025-11-28
> > >
> > > Thank you for your fast response, positive feedback, and support for our work with a higher rating.
> > >
> > > We are genuinely glad the concerns are resolved and appreciate all you do for the conference and research community.

---

### Official Review · Reviewer_1USz · 2025-11-01

**Soundness:** 3
**Presentation:** 3
**Contribution:** 3
**Rating:** 8
**Confidence:** 3

**Summary:**

This paper tackles controllable avatar generation that produces semantically meaningful, context-aware motion beyond basic lip-sync. The key challenge is aligning high-level cognitive intent with low-level reactive control, as naive multimodal fusion often causes semantic interference. To address this, it proposes a cognitive-simulation framework combining high-level reasoning (System 2) with low-level motion generation (System 1). Experiments demonstrate consistent improvements across multiple metrics.

**Strengths:**

This paper introduces a clear and creative dual-system framework (System 1 and System 2) for avatar generation, combining cognitive reasoning with generative modeling in a fresh way. The architecture includes well-designed and tested components such as the MMDiT backbone, semantic soft guidance, and the Pseudo Last Frame, which together improve the link between reasoning and motion generation. The experiments are thorough and the results show that the proposed method produces more expressive and context-aware avatars than strong baselines like VASA-1 and SadTalker, with ablation studies further supporting each design choice.

**Weaknesses:**

While MLLMs are powerful for reasoning, they come with high computational costs. For tasks such as analyzing a character's speech content, emotion, or planning shot composition, using a large MLLM may be unnecessary. A smaller, task-specific model jointly trained with the generative system could offer a more unified and efficient solution. Although incorporating an MLLM adds conceptual richness, the paper does not provide any quantitative analysis of the additional computational or latency overhead. Nonetheless, the work presents a creative and promising direction by integrating cognitive reasoning into avatar generation and demonstrates clear value. I would recommend a score above the acceptance bar.

**Questions:**

It would be helpful to better understand the robustness and efficiency of the proposed System 2 reasoning module. Specifically, are the results consistent when the reasoning outputs contain noise, particularly common hallucinations, or ambiguity in the input image or audio? Additionally, can the System 2 module be made more lightweight, and is there any analysis of its additional inference time or computational overhead?

---

> ### Author Response · Authors · 2025-11-19
>
> We sincerely thank the reviewer for their positive feedback on our work, including the methodology, contributions, experiments, and presentation. We are greatly encouraged by this recognition and will do our best to address the remaining concerns.
>
> > Regarding the Robustness and Efficiency of the System 2 Reasoning Module
>
> We sincerely thank the reviewer for their positive evaluation and insightful feedback. We agree that understanding the efficiency and robustness of the System 2 module is critical, and we are happy to provide further details on these points.
>
> 1. Computational Overhead and Latency
>
> We apologize for not making this information more prominent in the main paper. As detailed in Appendix F (page 30), the agentic reasoning module has a latency of approximately 20-30 seconds. This duration can fluctuate based on the target video length but does not increase linearly, as the reasoning process is executed in a single pass to generate the plan. This overhead is manageable and can be further optimized.
>
> 2. Potential for a More Lightweight System
>
> The reviewer correctly notes that a more lightweight system would be beneficial. Our agentic system is designed with modularity in mind, allowing for exactly this kind of optimization. Currently, as described in L910, we use miniCPM-o (an 8B MLLM) for the Analyser and Seed-1.5-VL (via API) for the Planner. This overhead can be reduced by:
>
> - Swapping Models: Replacing the current models with smaller, faster, or more unified MLLMs (e.g., Gemini Flash) as they become available.
> - Engineering Optimizations: Applying further acceleration techniques to the reasoning process.
>
> We also anticipate that the costs associated with API-based models will decrease over time as providers continue to optimize them, making this approach increasingly practical.
>
> 3. Robustness to Hallucinations and Input Ambiguity
>
> We acknowledge that hallucinations and consistency loss are challenges, particularly as generation duration and content dynamism increase. The system's robustness depends heavily on these factors:
>
> - In Low-Dynamism Scenarios: For static talking-head videos with repetitive speech, the focus of most current audio-driven avatar models, our model can generate coherent content for extended periods with minimal risk of hallucination.
> - In High-Dynamism Scenarios: For narratively complex content, performance limitations emerge. Our provided page includes a 90-second video with high dynamism (camera and character motion) that showcases our model's capabilities. In such cases, visual quality is maintained for about 2 minutes before artifacts like color shifting may appear. Narrative drift (hallucination) also becomes more likely when the script implies complex actions. For example, a line like "I'm about to board a plane" is handled reliably if the character only speaks, but if the expectation is a complex action sequence (e.g., pulling out a ticket, walking to security), the model is more prone to generating illogical behaviors.
>
> In summary, this is a known limitation. Our primary contribution is the introduction of a 'System 2' planning module, a novel and largely unexplored dimension in this field. We believe that the fundamental challenge of hallucination in generative models will be progressively addressed in future work, and our framework provides a new foundation for tackling it.

---

> > ### Comment · Reviewer_1USz · 2025-11-25
> >
> > Thank you for your efforts and your reply. My question has been resolved, and after reading the discussion among the other reviewers, I have decided to keep my original score.

---

> > > ### Author Response · Authors · 2025-11-28
> > >
> > > Thank you for your prompt and positive reply. I'm glad to know the question is resolved, and I truly appreciate your high-quality insights.

---

### Official Review · Reviewer_MK7E · 2025-11-11

**Soundness:** 3
**Presentation:** 3
**Contribution:** 3
**Rating:** 6
**Confidence:** 4

**Summary:**

This paper introduces a novel framework for generating dynamic and expressive video avatars by "instilling an active mind," drawing an analogy from the dual-process theory of human cognition (System 1 and System 2). The authors argue that existing avatar models are largely "System 1" agents, capable of reactive tasks like lip-sync but failing to exhibit higher-level semantics like emotion, intent, or contextual awareness. To bridge this gap, they propose a dual-system framework. A "System 2" module, termed Agentic Reasoning and powered by a Multimodal Large Language Model (MLLM), acts as a deliberative planner. It processes all input modalities (audio, reference image, text prompt) to generate a high-level, structured plan (in JSON format) that outlines the avatar's expressions, actions, and emotional journey. This plan then guides a "System 1" module, a reactive rendering engine built upon a novel Multimodal Diffusion Transformer (MMDiT). This engine synthesizes the final video, conditioned on both the high-level plan and the immediate audio signal. Key technical innovations include a "Pseudo Last Frame" (PLF) conditioning mechanism to maintain identity without sacrificing motion dynamics, and a specialized training strategy (MM-Branch Warm-up) to effectively train the complex multi-modal architecture. The authors demonstrate state-of-the-art performance through extensive experiments, including quantitative metrics and user studies, against a wide range of strong baselines.

**Strengths:**

1. The System 1/System 2 analogy is a standout feature, providing a clear and compelling motivation for the entire architecture. It reframes the problem in a way that is likely to inspire future work. Meanwhile, such method will improve the performance of avatars in the practice application.
2. The introduction of the Pseudo Last Frame (PLF) to decouple identity from static posture is a significant practical contribution that directly addresses a common failure mode in existing models.
3. The model achieves state-of-the-art performance across a comprehensive suite of metrics and, most tellingly, wins decisively in head-to-head user preference studies against top-tier competitors. And the paper provides textbook-quality ablation studies that convincingly demonstrate the necessity and effectiveness of its proposed components, from the high-level reasoning module down to specific architectural choices. Meanwhile, results shows that the model's success in diverse scenarios, including multi-person turn-taking and non-human avatars, highlights the robustness and power of the proposed framework.

**Weaknesses:**

1. About the Latentcy: The framework, involving a large MLLM for planning followed by a large diffusion transformer for rendering, is undoubtedly computationally intensive. The paper mentions training on a very large dataset (11,000 hours) and inference at 480p. A discussion of the computational requirements (e.g., VRAM, training time) and, more importantly, the ``inference latency'' would be valuable for understanding the practical deployability of the system.
2. The experiments primarily showcase clips of up to 720 frames (around 30 seconds). While the Agentic Reasoning module is designed for long-range planning, the paper does not explicitly test the limits of this coherence over much longer durations (e.g., several minutes). It would be interesting to know if any failure modes, such as narrative drift or loss of consistency, emerge in longer-form content generation.
3. A more complex settings with difficult reasoning will be beneficial to claim the improvement of this paper, and the setups of such benchmark will be useful for fairly comparison.

**Questions:**

See the Weakness.

---

> ### Author Response · Authors · 2025-11-19
>
> We sincerely thank the reviewer for their compliments on our designs for System 1 & 2 and for recognizing the value of our "pseudo last frame" approach. It is gratifying to have our work so well understood. We will now address your questions, aiming to clarify our work and improve your assessment of our paper.
>
> > Regarding Latency and Computational Requirements
>
> We thank the reviewer for this question. We apologize that this information was not more prominent.
> A detailed breakdown of computational costs is provided in Appendix B (page 17), as noted on L299. In summary, the training required:
> - Audio branch warm-up: ~18k GPU-hours (A100 equivalent)
> - Main training: ~43k GPU-hours
> - Fine-tuning: ~6k GPU-hours
>
> For inference latency, Appendix F (page 30) provides details. The 'Thinking' module requires 20-30 seconds. The video DiT, without optimization, takes approximately 15 minutes to generate a 10-second video. This inference time is common for current large-scale video DiT models.
>
> > Regarding Long-Form Content Generation and Coherence
>
> We thank the reviewer for this crucial point. We agree that long-range coherence is a key challenge.
> Many existing methods maintain stability by limiting content dynamism. In contrast, our provided page include a 90-second video with high dynamism (significant camera and character motion) to demonstrate our method's improved capabilities in this area. Our method maintains visual coherence for up to 2 minutes in such dynamic scenes, and significantly longer for static talking-head videos.
>
> However, we acknowledge that narrative drift and consistency loss remain challenges, especially when the script implies complex, multi-step actions. For instance, with a line like, "I'm about to board a plane," the model's performance depends on the expected output. If the character is only expected to speak the line, the model performs reliably. However, if the expectation is for the character to perform corresponding actions, such as pulling out a ticket and walking towards security, the model is more prone to generating incoherent or hallucinatory behaviors in these narratively complex, dynamic scenarios.
>
> This is a known limitation that we have not yet fully resolved. Our primary contribution is the introduction of a 'System 2' agentic planning module, a largely unexplored dimension in avatar generation. While long-term consistency is a critical issue, we see it as a broader research challenge for the field that our work provides a new foundation to build upon.
>
> > Regarding Evaluation on More Complex Reasoning Tasks
>
> We agree with the reviewer on the need for benchmarks that test complex reasoning. A key challenge is the current lack of such benchmarks, as most existing evaluations focus on 'System 1' tasks (e.g., lip-sync) rather than the 'System 2' reasoning our work introduces.
>
> Given this gap, we introduced our own metrics to assess motion rationality. Our results for 'Motion Unnaturalness' (Table 2) and 'Motion-Context Coherence' (Table 5) were designed to specifically measure the benefits of our reasoning module and demonstrate a clear improvement over baseline approaches.
> We concur this is a critical direction for future work. A robust benchmark should include tasks of varying difficulty and employ a combination of subjective user studies (with trained participants) and objective LLM-assisted evaluations to properly assess reasoning quality.

---

### Author Response · Authors · 2025-12-01
**Brief Summary of the Author-Reviewer Discussion**

Dear AC, SAC, and PC,

Thank you for your time and for managing the review process for the conference. We understand that this year's circumstances may have increased the reviewing load, and we hope a brief summary of the author-reviewer discussion might be helpful.

First and foremost, we are deeply grateful for the thoughtful feedback from all reviewers. We are very encouraged that they recognized the value of our work, particularly the novelty of introducing a 'System 2' planning module to the avatar generation field.

Following the discussion period, we are pleased to report the following positive outcomes:

1.  **Reviewers 1USz (Score: 8) and fLxy (Score: 8)** both confirmed that their concerns have been fully addressed. We provided detailed clarifications on the MLLM's computational overhead and its technical synergy with the rendering model, and they kindly expressed their satisfaction with our responses.

2.  **Reviewer Rgov (Score: 6, expressed intent to raise)** also confirmed that their questions were resolved. They explicitly stated their intention to raise the score but were unable to do so due to system limitations. Their primary questions were about the framework's novelty and the specifics of our PLF mechanism, which we clarified with quantitative results and detailed explanations.

3.  **Reviewer MK7E (Score: 6)** was unable to provide a final comment before the discussion period closed. We provided a detailed response to their questions regarding inference latency, long-form content coherence, and the need for complex reasoning benchmarks. Our response aimed to be thorough: we not only pointed to existing details in the appendix but also provided further clarifications. Moreover, we elaborated on our core contribution and discussed future directions for benchmarking. Our reply was quite comprehensive, and as such, we are optimistic that it would address their concerns.

In summary, we are delighted with the overwhelmingly positive reception from the reviewers and their appreciation for this new paradigm. **We sincerely believe that our work, representing a pioneering effort to integrate 'System 1' and 'System 2' functionalities in an analogous way, offers valuable insights for the avatar research community.**

We sincerely thank you and the entire reviewing team for your valuable contributions to the ICLR community.

Best regards,

The Authors of Submission2569

---

### Meta-Review · Area_Chair_u8Ht · 2026-01-10

**Summary:**

All reviewers provided positive initial ratings, so it would be a clear accept.

**Reviewer Concerns:**

1. Computational & latency overhead (raised by 1USz, MK7e, Rgov)

– Authors now transparently report 20-30 s “thinking” time + ≈ 15 min for 10 s 480 p video, place full tables in appendices, and outline future paths (smaller MLLMs, distillation).

2. Long-form coherence / narrative drift (MK7e, 1USz)

– 90 s demo supplied; artefacts appear after ~2 min in highly dynamic scenes. Authors candidly label this an open problem; reviewers accepted the limitation because (a) it is already better than baselines and (b) the paper’s core contribution is the introduction of a System-2 planner, not a final solution to long-range consistency.

3. Novelty of “cognitive simulation” vs. existing planner–executor literature (Rgov)

– Authors clarified that while the high-level structure is known, its non-trivial adaptation to audio-driven avatar generation (multimodal persona & emotion reasoning, PLF, MM-DiT training strategy) yields measurable improvements (+28 % GSB on motion-context coherence).

4. Robustness to MLLM hallucinations / noisy plans (1USz)

– Empirical ablations show graceful degradation; low-dynamism talking-head clips remain stable.

5. Need for richer System-2 benchmarks (MK7e)

– Authors acknowledge the gap, release new metrics (Motion Unnaturalness, Motion-Context Coherence), and commit to sharing code/data.

**Reviewer Scores:**

N.A.

---

### Decision · Program_Chairs · 2026-01-26

Accept (Oral)